# LABEL-FREE GUI GROUNDING VIA CONFIDENCE-GUIDED NEGATIVE REINFORCEMENT LEARNING

## ABSTRACT

Graphical User Interface (GUI) grounding maps natural language instructions to precise interface locations for autonomous interaction. Current supervised fine-tuning and reinforcement learning approaches rely heavily on costly annotated data, creating a bottleneck for scaling GUI agents. We introduce a label-free training paradigm leveraging two key insights: (1) coordinate tokens in model outputs exhibit distinct confidence patterns that reliably identify correct predictions, and (2) in sparse GUI coordinate spaces, negative samples provide more reliable learning signals than potentially corrupted positive ones. We propose Confidence-Guided Reinforcement Learning (CRL), which uses coordinate-token confidence to select pseudo-labels from multiple samples and assigns distance-based rewards. We further develop Confidence-Guided Negative Reinforcement Learning (CNRL), which exclusively learns from negative samples. Without using any annotations, CNRL-7B achieves 92.1% on ScreenSpot-V2, surpassing UI-TARS-72B (90.3%) trained on 18.4M labels. On ScreenSpot-Pro, CNRL-7B reaches 33.8%, improving 8.9% absolute over the base model and exceeding GUI-R1-7B (31.0%) trained on 3K labels. On challenging high-resolution benchmarks, CNRL consistently outperforms CRL by 1-1.5%, demonstrating that learning what to avoid can be more effective than learning from uncertain positive examples. Our findings establish coordinate-token confidence as a powerful alternative to manual annotations for scalable GUI agent development.

## 1 INTRODUCTION

GUI agents have emerged as a critical technology for automating human-computer interaction, enabling natural language commands to be translated into precise interface actions. While early rule-based systems required extensive manual engineering and lacked adaptability (Zhou et al., 2024a; Wang et al., 2024a), the advent of multimodal large language models (MLLMs) (Bai et al., 2023; OpenAI, 2024; Bai et al., 2025a; Zhou et al., 2024b) has opened new possibilities for flexible GUI understanding and interaction (Tang et al., 2025c; Hu et al., 2025). At the heart of these agents lies GUI grounding, the task of mapping natural language instructions to specific interface elements through coordinate prediction (Cheng et al., 2024; Lin et al., 2025; Wu et al., 2024a).

Current approaches to GUI grounding employ two main training paradigms, as illustrated in Figure 1: (1) Supervised Fine-Tuning (SFT) (Hong et al., 2024; Cheng et al., 2024; Gou et al., 2024; Wu et al., 2024b; Lin et al., 2025; Tang et al., 2025b; Xu et al., 2024; Qin et al., 2025; Wu et al., 2025; Sun et al., 2025), which directly learns from ground-truth bounding box annotations, and (2) Reinforcement Learning with Verifiable Rewards (RLVR) (Lu et al., 2025a; Luo et al., 2025; Liu et al., 2025a; Zhou et al., 2025; Yuan et al., 2025; Tang et al., 2025d; Liu et al., 2025b; Lu et al., 2025b), which requires ground-truth labels to compute accurate rewards for policy optimization. However, both paradigms face a fundamental challenge: the heavy reliance on annotated data. Obtaining pixel-level bounding box annotations is prohibitively expensive and time-consuming, requiring manual labeling of precise coordinates for each UI element. This dependency on labeled data severely limits the scalability and practical deployment of GUI agents across diverse applications and domains.

A natural question arises: can we enhance GUI grounding capabilities without explicit supervision? Test-time reinforcement learning (TTRL) (Zuo et al., 2025) offers a promising direction, having

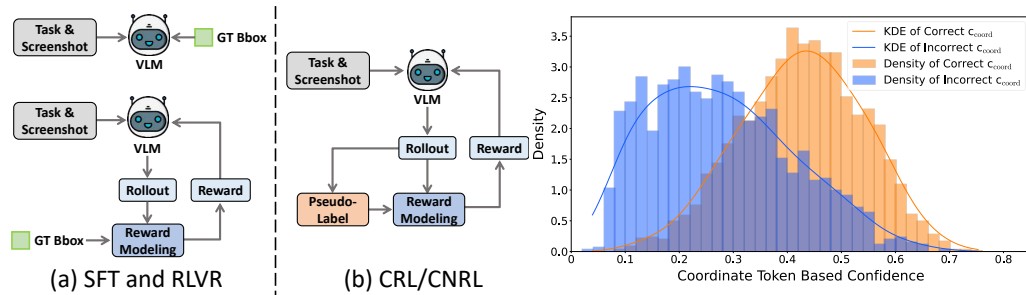

Figure 1: Training paradigms for GUI grounding and validation of coordinate-token confidence. **Left**: Comparison of training methods: (a) SFT and RLVR require ground truth labels, while (b) our CRL/CNRL methods leverage this confidence signal for label-free training. **Right**: Distribution of coordinate-token confidence on ScreenSpot-V2 shows clear separation between correct and incorrect predictions, with correct predictions exhibiting significantly higher confidence values, validating its effectiveness as a reliability indicator.

demonstrated success in mathematical reasoning tasks through label-free adaptation. However, applying TTRL to GUI grounding presents unique challenges. Unlike mathematical problems where correctness can be verified through symbolic computation, GUI tasks require spatial reasoning over visual elements where ground truth is typically unavailable during inference.

To address this challenge, we propose leveraging the model's own confidence signals as a source of supervision. Our key insight is that while a model's most confident predictions may not always be correct, its coordinate-level token probabilities provide valuable signals for distinguishing likely correct from incorrect predictions. As illustrated in Figure 1, we empirically observe that the confidence distributions of correct and incorrect predictions form distinct, well-separated clusters when focusing on coordinate tokens. This observation motivates our development of coordinate-token confidence, a metric that focuses specifically on the probability values of tokens representing spatial coordinates rather than the entire response sequence.

Building on this foundation, we introduce **C**onfidence-Guided **R**einforcement **L**earning (CRL), which generates multiple candidate predictions, selects the most confident one as a pseudo-label based on coordinate-token confidence, and performs reinforcement learning using binary rewards derived from spatial distances to this pseudo-label. As shown in Figure 1(b), our approach eliminates the need for ground truth labels by using these pseudo-labels to assign rewards directly, enabling truly label-free training. However, pseudo-labels inherently contain errors that could negatively impact learning. This leads to a critical observation: in the sparse coordinate space of GUI grounding, negative samples (predictions far from the pseudo-label) are overwhelmingly likely to be incorrect, while positive samples near potentially misplaced pseudo-labels may be unreliable. This asymmetry motivates **C**onfidence-Guided **N**egative **R**einforcement **L**earning (CNRL), which modifies the advantage computation to zero out potentially unreliable positive samples while preserving negative learning signals. By focusing solely on reliably incorrect predictions, CNRL transforms the challenge of pseudo-label uncertainty into an opportunity for robust learning.

Extensive evaluation across four benchmarks validates our approach. Without any annotations, CRL-7B achieves 92.1% on ScreenSpot-V2, surpassing UI-TARS-72B (90.3%) trained on 18.4M labels. CNRL-7B reaches 33.8% on ScreenSpot-Pro, exceeding GUI-R1-7B (31.0%) which requires ground truth rewards. On challenging high-resolution benchmarks, CNRL consistently outperforms CRL by 1-1.5%, confirming that negative samples provide more reliable supervision when targets are small and sparse. Analysis reveals coordinate-token confidence outperforms eight alternative pseudo-labeling strategies by 2.1-11.9%, while CNRL maintains stable performance across predefined distance thresholds compared to CRL's variance, demonstrating superior robustness for practical deployment.

Our contributions can be summarized as follows:

- We propose CRL, a label-free training approach for GUI grounding that introduces coordinate-token confidence to identify optimal predictions among multiple candidates and uses these as pseudo-labels for reinforcement learning without manual annotations.

- We further develop CNRL, which exclusively uses negative samples during reinforcement learning, demonstrating that learning from reliably incorrect predictions can be more effective than using potentially inaccurate positive samples in label-free settings.

- Extensive experiments demonstrate that our label-free approaches achieve competitive performance across benchmarks like ScreenSpot-V2, ScreenSpot-Pro and UI-Vision.

## 2 RELATED WORK

### 2.1 GUI GROUNDING

GUI grounding bridges natural language instructions with GUI elements, enabling agents to understand and interact with software environments through grounded multimodal reasoning (Cheng et al., 2024). Given a screenshot and a natural language command, the task requires identifying the corresponding UI element and returning its location as either a bounding box or a point coordinate. Early efforts in enhancing GUI grounding capabilities primarily relied on supervised fine-tuning (SFT), where models like SeeClick (Cheng et al., 2024), UGround (Gou et al., 2024), CogAgent (Hong et al., 2024), OS-ATLAS (Wu et al., 2024b) and Aguvis (Xu et al., 2024) were trained on large-scale annotated datasets. Following the introduction of Group Relative Policy Optimization (GRPO) (Shao et al., 2024) and DeepSeek-R1 (Guo et al., 2025), recent work has shifted toward RLVR, where models such as UI-R1 (Lu et al., 2025a), GUI-R1 (Luo et al., 2025), InfiGUI-R1 (Liu et al., 2025a), GUI-G1 (Zhou et al., 2025), SE-GUI (Yuan et al., 2025), and GUI-G$^2$ (Tang et al., 2025a) leverage interaction signals and feedback to further enhance grounding performance. However, both SFT and RLVR paradigms fundamentally depend on labeled data, where manual annotation is prohibitively expensive and automated labeling processes often introduce significant errors, creating a major bottleneck for scaling GUI agents to diverse applications and domains.

### 2.2 NEGATIVE LEARNING

Negative learning (NL) reframes supervised learning by training models on what to avoid rather than what to produce. It operates on the principle that a model's least confident predictions serve as reliable negative signals, even when its most confident predictions may be incorrect (Kim et al., 2019). This approach has proven effective for tasks with noisy labels and in few-shot settings where robust training signals are scarce (Wei et al., 2022). Recently, NL has been successfully applied to large language models for mathematical reasoning, using either an increased ratio of negative samples (Chen et al., 2025; Wang et al., 2024b) or negative samples exclusively (Zhu et al., 2025). Despite its success in text-based domains, the application of NL to multimodal models remains unexplored. Our work is the first to investigate whether learning solely from negative examples can be an effective strategy for complex vision-language tasks such as GUI grounding.

## 3 METHOD

We propose a label-free training paradigm for GUI grounding. Our approach consists of two key components: (1) coordinate-token confidence-based pseudo-label generation that selects the most reliable prediction from multiple samples, and (2) distance-based reinforcement learning that assigns rewards without ground truth. We present two variants: **C**onfidence-Guided **R**einforcement **L**earning (CRL) that performs policy optimization using both positive and negative rewards, and **C**onfidence-Guided **N**egative **R**einforcement **L**earning (CNRL) that exclusively learns from negative samples to mitigate pseudo-label errors in the GUI coordinate space.

### 3.1 CONFIDENCE-BASED PSEUDO-LABEL GENERATION

The core challenge in label-free GUI grounding is identifying reliable training signals from the model's own predictions. Standard confidence estimation averages probabilities across all tokens:

$$c(y \mid x) = \frac{1}{n} \sum_{i=1}^{n} P(y_i \mid y_1, \dots, y_{i-1}, x) \quad (1)$$

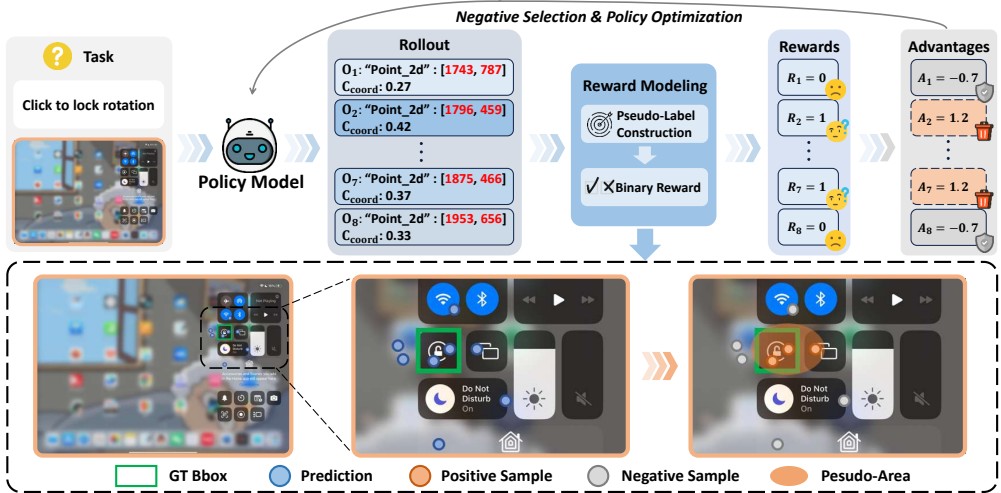

Figure 2: The CNRL pipeline. Given an instruction and screenshot, the model generates multiple predictions with associated coordinate-token confidence scores, selects the highest-confidence prediction as a pseudo-label, and assigns binary rewards based on distance threshold $\tau$. During policy optimization, CNRL leverages only negative samples (gray) while zeroing advantages for potentially unreliable positive samples (orange), as illustrated in the bottom panel showing the transformation from standard to negative-only reinforcement learning.

However, this approach fails in GUI tasks because formatting and descriptive tokens consistently exhibit high probabilities regardless of prediction quality, masking uncertainty in the critical coordinate values. Our analysis reveals that coordinate tokens—the numerical values specifying pixel locations—carry the primary uncertainty signal, with 71.5% having probabilities below 0.6 while 78.1% of non-coordinate tokens exceed 0.9 on ScreenSpot-V2. We therefore introduce coordinate-token confidence, which focuses exclusively on these informative tokens:

$$c_{\text{coord}}(y \mid x_{img}, x_{ins}) = \frac{1}{k} \sum_{i \in \text{coord}} P(y_i \mid y_1, \ldots, y_{i-1}, x_{img}, x_{ins}) \tag{2}$$

where $k$ represents the number of coordinate tokens. As shown in Figure 1 right, this metric effectively distinguishes correct from incorrect predictions. Given an image–instruction pair $(x_{img}, x_{ins})$, we generate multiple predictions through parallel sampling and select the one with highest coordinate-token confidence as our pseudo-label:

$$\hat{y} = \arg\max_{y \in \mathcal{Y}} c_{\text{coord}}(y \mid x_{img}, x_{ins}), \quad \hat{p} = \text{extract}(\hat{y}) \tag{3}$$

where $\hat{y}$ represents the optimal response selected by coordinate-token confidence, and $\mathcal{Y}$ denotes the collection consisting of generated responses. The function extract($y$) is defined to obtain a point from the response $y$. $\hat{p}$ indicates the optimal predicted point, i.e. the psudo-label. This confidence-based selection provides a principled basis for self-supervised learning without requiring ground truth annotations.

## 3.2 CONFIDENCE-GUIDED REINFORCEMENT LEARNING

To enable reinforcement learning without ground truth annotations, we design a distance-based reward mechanism using our pseudo-labels. For the $i$th sampled prediction $(x_i, y_i)$, given an image with height $H$ and width $W$, we normalize its coordinates as $(x_i/W, y_i/H)$, and compute its Euclidean distance to the pseudo-label and assign binary rewards:

$$R_i = \begin{cases} 1, & \text{if } \text{dist}(p_i, \hat{p}) \leq \tau \\ 0, & \text{otherwise} \end{cases} \tag{4}$$

This formulation exploits the sparse nature of GUI coordinate space where predictions close to our high-confidence pseudo-label are likely correct, while distant predictions are almost certainly wrong.

The distance threshold $\tau$ defines the boundary between potentially correct and incorrect predictions. We then apply Group Relative Policy Optimization (GRPO) (Shao et al., 2024), which estimates advantages through standardization across multiple samples:

$$A_i = \frac{R_i - \text{mean}(\{R_j\}_{j=1}^N)}{\text{std}(\{R_j\}_{j=1}^N)} \tag{5}$$

where $N$ donates the sampling number. The policy optimization follows the standard GRPO objective with clipped probability ratios and KL regularization:

$$\mathcal{J}_{\text{GRPO}}(\theta) = \mathbb{E}_{q,o_i} \frac{1}{G} \sum_{i=1}^G \left[ \min\left(r_i(\theta)A_i, \text{clip}(r_i(\theta), 1-\epsilon, 1+\epsilon)A_i\right) - \beta \mathbb{D}_{\text{KL}}[\pi_\theta | \pi_{\text{ref}}] \right] \tag{6}$$

where the probability ratio $r_i(\theta)$ prevents excessive policy updates and $\beta$ controls the strength of regularization. By treating the model's most confident prediction as a learning target and using spatial proximity to define rewards, CRL transforms GUI grounding into a self-supervised reinforcement learning problem requiring no external annotations.

### 3.3 CONFIDENCE-GUIDED NEGATIVE REINFORCEMENT LEARNING

While CRL provides a path to label-free training, pseudo-labels inevitably contain errors that can mislead learning, particularly when incorrect predictions are mistakenly rewarded as positive examples. However, the sparse nature of GUI coordinate space offers an opportunity: negative samples are overwhelmingly reliable since predictions far from any reasonable target are almost certainly incorrect. CNRL exploits this asymmetry by modifying the advantage computation to use only negative samples:

$$A_i = \begin{cases} 0, & \text{if } R_i = 1 \\ \frac{-\text{mean}(\{R_j\}_{j=1}^N)}{\text{std}(\{R_j\}_{j=1}^N)}, & \text{if } R_i = 0 \end{cases} \tag{7}$$

This approach zeros out advantages for potentially unreliable positive samples while preserving the learning signal from negative samples. The model thus learns exclusively from what to avoid rather than what to produce, which proves particularly effective in high-resolution GUI tasks where the vast coordinate space makes distant predictions almost certainly incorrect. As illustrated in Figure 2, CNRL maintains the same pseudo-label generation and reward assignment pipeline as CRL but selectively updates the model using only the most reliable learning signals. This strategy effectively converts the challenge of label uncertainty into an opportunity for robust negative learning, achieving superior performance on challenging datasets where pseudo-label quality may be compromised.

## 4 EXPERIMENTS

### 4.1 EXPERIMENTS SETUP

**Datasets and Benchmarks.** We evaluate on four benchmarks: ScreenSpot (Cheng et al., 2024) and ScreenSpot-V2 (Wu et al., 2024b), which have low-resolution interfaces and larger targets, and more challenging benchmark ScreenSpot-Pro (Li et al., 2025) and UI-Vision (Nayak et al., 2025). Predictions are correct if they fall within ground truth bounding boxes.

**Implementation Details**. We use Qwen-2.5-VL (Bai et al., 2025b) (3B/7B) as the base model within the VLM-R1 framework (Shen et al., 2025). We conduct training and testing on each benchmark individually, and we do not split the training and test set. For UI-Vision, we only use the Element Grounding subset for training and evaluation, without Layout Grounding and Action Prediction subset. All training are conducted for 1 epoch, with learning rate 1e-6. To enhance diversity during sampling, we set temperature $T = 1.0$, $top\_k = 50$, $top\_p = 1.0$. KL penalty $\beta$ is set to 0.04. Distance threshold $\tau$ is set to 0.05. We apply Flash Attention2 (Dao, 2023) for training. Following Zuo et al. (2025), to reduce computational costs while improve the accuracy of pseudo-labels, we apply downsampling after pseudo-label construction for optimze policy model. Specifically, we sample 16 responses to construct pseudo labels. For CRL, we randomly downsample 8 responses to compute the advantages. For CNRL, we compute the advantages using 16 responses and then select the 8 negative responses farthest from the pseudo-label, filling in with positive responses if fewer

Table 1: Performance comparison on ScreenSpot and ScreenSpot-V2. "-" indicates missing values due to unavailable results in the original paper, unreleased model checkpoints, and inference code.

| Model | Labels | v1 Mobile | | v1 Desktop | | v1 Web | | v1 Avg. | v2 Avg. |
|---|---|---|---|---|---|---|---|---|---|
| | | Text | Icon | Text | Icon | Text | Icon | | |
| *Proprietary Models* | | | | | | | | | |
| GPT-4o | - | 30.5 | 23.2 | 20.6 | 19.4 | 11.1 | 7.8 | 18.8 | 20.1 |
| Claude Computer Use | - | - | - | - | - | - | - | 83.0 | - |
| *GUI-specific Models (SFT)* | | | | | | | | | |
| CogAgent-18B | 222M | 67.0 | 24.0 | 74.2 | 20.0 | 70.4 | 28.6 | 47.4 | - |
| SeeClick-9.6B | 1M | 78.0 | 52.0 | 72.2 | 30.0 | 55.7 | 32.5 | 53.4 | 55.1 |
| UGround-7B | 10M | 82.8 | 60.3 | 82.5 | 63.6 | 80.4 | 70.4 | 73.3 | 76.3 |
| OS-Atlas-7B | 13M | 93.0 | 72.9 | 91.8 | 62.9 | 90.9 | 74.3 | 82.5 | - |
| ShowUI-2B | 256K | 92.3 | 75.5 | 76.3 | 61.1 | 81.7 | 63.6 | 75.1 | 77.3 |
| Focus-2B | 300K | 90.1 | 78.2 | 80.9 | 65.0 | 81.7 | 68.5 | 77.4 | - |
| Aguvis-7B | 1M | 95.6 | 77.7 | 93.8 | 67.1 | 88.3 | 75.2 | 84.4 | 80.5 |
| Aguvis-72B | 1M | 94.5 | 85.2 | 95.4 | 77.9 | 91.3 | **85.9** | 89.2 | - |
| UI-TARS-7B | 18.4M | 94.5 | 85.2 | 95.9 | 85.7 | 90.0 | 83.5 | 89.5 | 91.6 |
| UI-TARS-72B | 18.4M | 94.9 | 82.5 | 89.7 | **88.6** | 88.7 | 85.0 | 88.4 | 90.3 |
| GUI-Actor-7B | 9.6M | 94.9 | 82.1 | 91.8 | 80.0 | 91.3 | 85.4 | 88.3 | **92.1** |
| Jedi-3B | 4M | - | - | - | - | - | - | - | 88.6 |
| Jedi-7B | 4M | - | - | - | - | - | - | - | 91.7 |
| *GUI-specific Models (RL)* | | | | | | | | | |
| UI-R1-3B | 136 | 95.6 | 84.7 | 90.2 | 59.3 | 85.2 | 73.3 | 83.3 | 85.4 |
| UI-R1-E-3B | 2K | 97.1 | 83.0 | 95.4 | 77.9 | **91.7** | 85.0 | 89.2 | 89.5 |
| GUI-R1-3B | 3K | - | - | 93.8 | 64.8 | 89.6 | 72.1 | - | - |
| GUI-R1-7B | 3K | - | - | 91.8 | 73.6 | 91.3 | 75.7 | - | - |
| InfiGUI-R1-3B | 32K | 97.1 | 81.2 | 94.3 | 77.1 | **91.7** | 77.6 | 87.5 | - |
| GUI-G1-3B | 17K | **98.6** | **85.8** | **96.4** | 80.7 | 91.4 | 82.3 | **90.3** | - |
| SE-GUI-7B | 3K | - | - | - | - | - | - | 88.2 | 90.3 |
| *Vanilla Models* | | | | | | | | | |
| Qwen-2.5-VL-3B | 0 | 93.8 | 68.1 | 91.2 | 55.0 | 81.7 | 64.6 | 77.6 | 82.1 |
| Qwen-2.5-VL-7B | 0 | 91.9 | 80.8 | 88.1 | 75.7 | 90.0 | 77.7 | 84.9 | 88.1 |
| *Ours* | | | | | | | | | |
| CRL-3B | 0 | 96.7 | 78.6 | 95.4 | 67.9 | 87.8 | 72.8 | 84.6 | 88.9 |
| CNRL-3B | 0 | 96.0 | 79.0 | **96.4** | 66.4 | 87.0 | 73.8 | 84.5 | 88.5 |
| CRL-7B | 0 | **97.1** | **87.3** | 86.1 | 80.7 | **91.7** | 83.0 | 88.6 | **92.1** |
| CNRL-7B | 0 | 96.7 | **87.3** | 88.6 | **82.1** | 91.3 | **84.0** | **89.2** | **92.1** |

than 8 negatives are available. Since the advantages of positive responses are set to 0 in CNRL, they do not affect the computation. More details of hyperparameters can be found in Appendix A.11.

**Baselines.** We compare against three categories of methods: (1) SFT models including SeeClick (Cheng et al., 2024), ShowUI (Lin et al., 2025), UGround (Gou et al., 2024), OS-Atlas (Wu et al., 2024b), and UI-TARS (Qin et al., 2025) trained on datasets ranging from 256K to 18.4M labeled examples; (2) Reinforcement learning methods like UI-R1 (Lu et al., 2025a), GUI-R1 (Luo et al., 2025), and InfiGUI-R1 (Liu et al., 2025a) that require ground truth for reward computation; (3) Proprietary models GPT-4o (OpenAI, 2024) and Claude Computer Use (Anthropic, 2024).

## 4.2 MAIN RESULTS

**Our label-free methods achieve competitive performance without any annotations.** Tables 1 and 2 demonstrate substantial improvements over vanilla base models across all benchmarks, with absolute gains ranging from 4.0% to 8.9%. On ScreenSpot-V2, CNRL-7B achieves 92.1%, improving 4.0% over the base Qwen-2.5-VL-7B, matching GUI-Actor-7B (92.1%) which uses 9.6M labels, and surpassing ShowUI-2B (77.3%) by 14.8% despite using zero annotations. Compared to RL baselines, our approach shows clear advantages: CNRL-7B outperforms GUI-R1-7B by 3.1% on

Table 2: Performance comparison on ScreenSpot-Pro and UI-Vision. Pro Avg. and UI-V Avg. represent overall accuracy across all categories for each respective benchmark.

| Model | Labels | CAD | | Dev | | Creative | | Scientific | | Office | | OS | | Pro Avg. | UI-V Avg. |
|---|---|---|---|---|---|---|---|---|---|---|---|---|---|---|---|
| | | Text | Icon | Text | Icon | Text | Icon | Text | Icon | Text | Icon | Text | Icon | | |
| *Proprietary Models* | | | | | | | | | | | | | | | |
| GPT-4o | - | 2.0 | 0.0 | 1.3 | 0.0 | 1.0 | 0.0 | 2.1 | 0.0 | 1.1 | 0.0 | 0.0 | 0.0 | 0.8 | 1.4 |
| Claude Computer Use | - | 14.5 | 3.7 | 22.0 | 3.9 | 25.9 | 3.4 | 33.9 | 15.8 | 30.1 | 16.3 | 11.0 | 4.5 | 17.1 | 8.3 |
| *GUI-specific Models (SFT)* | | | | | | | | | | | | | | | |
| SeeClick-9.6B | 1M | 2.5 | 0.0 | 0.6 | 0.0 | 1.0 | 0.0 | 3.5 | 0.0 | 1.1 | 0.0 | 2.8 | 0.0 | 1.1 | 5.4 |
| Focus-2B | 300K | 7.6 | 3.1 | 22.8 | 1.7 | 23.7 | 1.7 | 25.0 | 7.1 | 23.2 | 7.7 | 17.8 | 2.5 | 13.3 | - |
| CogAgent-18B | 222M | 7.1 | 3.1 | 14.9 | 0.7 | 9.6 | 0.0 | 22.2 | 1.8 | 13.0 | 0.0 | 5.6 | 0.0 | 7.7 | - |
| Aria-UI | 17.6M | 7.6 | 1.6 | 16.2 | 0.0 | 23.7 | 2.1 | 27.1 | 6.4 | 20.3 | 1.9 | 4.7 | 0.0 | 11.3 | 10.1 |
| OS-Atlas-7B | 13M | 12.2 | 4.7 | 33.1 | 1.4 | 28.8 | 2.8 | 37.5 | 7.3 | 33.9 | 5.7 | 27.1 | 4.5 | 18.9 | 9.0 |
| ShowUI-2B | 256K | 2.5 | 0.0 | 16.9 | 1.4 | 9.1 | 0.0 | 13.2 | 7.3 | 15.3 | 7.5 | 10.3 | 2.2 | 7.7 | 5.9 |
| UGround-7B | 10M | 14.2 | 1.6 | 26.6 | 2.1 | 27.3 | 2.8 | 31.9 | 2.7 | 31.6 | 11.3 | 17.8 | 0.0 | 16.5 | - |
| UGround-v1-7B | - | 15.8 | 1.2 | 51.9 | 2.8 | 47.5 | 9.7 | 57.6 | 14.5 | 60.5 | 13.2 | 38.3 | 7.9 | 31.1 | 12.9 |
| UI-TARS-7B | 18.4M | 20.8 | 9.4 | 58.4 | 12.4 | 50.0 | 9.1 | 63.9 | **31.8** | 63.3 | 20.8 | 30.8 | **16.9** | 35.7 | **17.6** |
| Jedi-3B | 4M | 27.4 | 9.4 | **61.0** | **13.8** | **53.5** | 8.4 | 54.2 | 18.2 | 64.4 | 32.1 | 38.3 | 9.0 | 36.1 | - |
| Jedi-7B | 4M | **38.0** | 14.1 | 42.9 | 11.0 | 50.0 | **11.9** | **72.9** | 25.5 | **75.1** | **47.2** | 33.6 | **16.9** | **39.5** | - |
| ZonUI-3B | 24K | 31.9 | **15.6** | 24.6 | 6.2 | 40.9 | 7.6 | 54.8 | 18.1 | 57.0 | 26.4 | 19.6 | 7.8 | 28.7 | - |
| *GUI-specific Models (RL)* | | | | | | | | | | | | | | | |
| InfiGUI-R1-3B | 32K | 33.0 | 14.1 | 51.3 | 12.4 | 44.9 | 7.0 | 58.3 | 20.0 | 65.5 | 28.3 | **43.9** | 12.4 | 35.7 | - |
| UI-R1-E-3B | 2K | 37.1 | 12.5 | 46.1 | 6.9 | 41.9 | 4.2 | 56.9 | 21.8 | 65.0 | 26.4 | 32.7 | 10.1 | 33.5 | - |
| GUI-R1-7B | 3K | 23.9 | 6.3 | 49.4 | 4.8 | 38.9 | 8.4 | 55.6 | 11.8 | 58.7 | 26.4 | 42.1 | **16.9** | 31.0 | - |
| *Vanilla Models* | | | | | | | | | | | | | | | |
| Qwen-2.5-VL-3B | 0 | 9.1 | 7.3 | 22.1 | 1.4 | 26.8 | 2.1 | 38.2 | 7.3 | 33.9 | 15.1 | 10.3 | 1.1 | 16.1 | 12.0 |
| Qwen-2.5-VL-7B | 0 | 13.7 | 7.8 | 44.2 | 6.9 | 28.8 | 10.5 | 48.6 | 5.5 | 46.9 | 15.1 | 31.8 | 12.4 | 24.9 | 15.0 |
| *Ours* | | | | | | | | | | | | | | | |
| CRL-3B | 0 | 35.0 | 9.4 | 44.2 | 3.4 | **46.5** | 4.9 | 60.4 | 18.1 | 53.1 | 20.8 | 38.3 | 7.9 | 32.1 | 17.2 |
| CNRL-3B | 0 | **39.1** | 7.8 | 42.9 | 4.8 | 46.0 | 3.5 | 58.3 | **20.9** | 55.9 | 22.6 | 38.3 | 7.9 | 32.7 | 18.5 |
| CRL-7B | 0 | 25.4 | **11.0** | 51.3 | 6.9 | 38.9 | **9.8** | 60.4 | 15.5 | **64.7** | 19.2 | 37.4 | **19.2** | 32.7 | 18.6 |
| CNRL-7B | 0 | 29.4 | 10.9 | **53.2** | **9.0** | 37.4 | 8.4 | **63.2** | 14.5 | 62.1 | **26.4** | **40.2** | 15.7 | **33.8** | **20.1** |

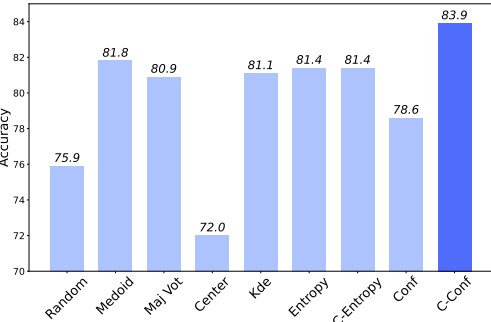

Figure 3: Comparison of pseudo-label construction methods with sampling number $N$=12.

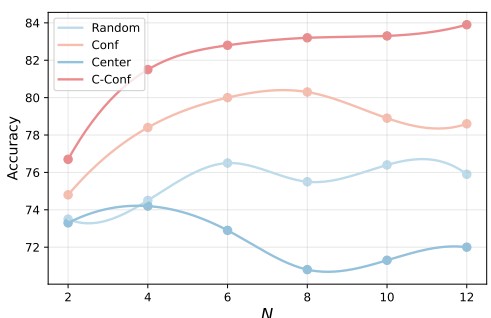

Figure 4: Pseudo-label quality scales with sampling budget across four strategies.

ScreenSpot while eliminating the need for 3K ground truth labels. At the 3B scale, CNRL improves the base model from 77.6% to 84.5% (+6.9%), approaching InfiGUI-R1-3B (87.5%) which requires 32K labels, validating that coordinate-token confidence provides sufficient supervision.

**Negative learning dominates on challenging high-resolution benchmarks.** The performance gap between CRL and CNRL reveals a critical pattern: as difficulty increases, exclusive negative learning becomes increasingly advantageous. On ScreenSpot-Pro, CNRL-7B achieves 33.8%, improving 1.1% over CRL-7B and 8.9% over the base model. This advantage extends to UI-Vision where CNRL-7B reaches 20.1%, surpassing CRL by 1.5% and dramatically exceeding supervised baselines like UI-TARS-7B (17.6%). The systematic superiority of CNRL on high-resolution tasks confirms that when targets occupy <1% of screen area, negative samples provide more reliable learning signals than potentially incorrect positive samples derived from pseudo-labels.

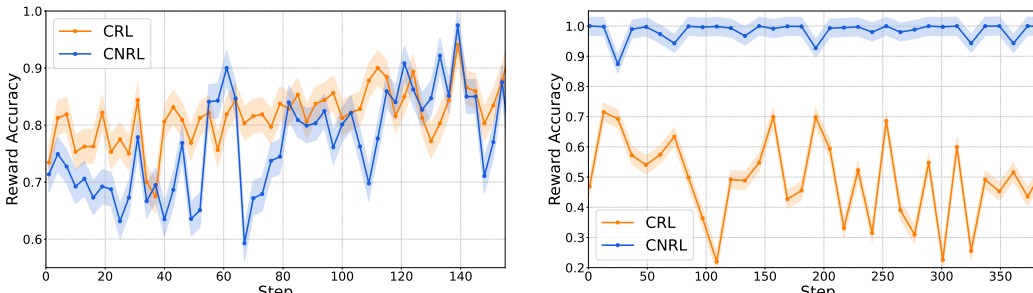

Figure 5: **Left**: Reward accuracy on ScreenSpot-V2. **Right**: Reward accuracy on ScreenSpot-Pro.

## 4.3 IN-DEPTH ANALYSIS

**Coordinate-token confidence significantly outperforms alternative pseudo-labeling strategies.**
Figure 3 evaluates nine different pseudo-label selection methods on ScreenSpot-V2 without training. Our proposed coordinate-token confidence (C-Conf) achieves 83.9% accuracy with N=12 samples, substantially exceeding all alternatives including standard all-token confidence (78.6%), spatial clustering methods like KDE (81.1%) and majority voting (80.9%), and entropy-based approaches (81.4%). The 5.3% improvement over standard confidence validates our core insight that coordinate tokens carry the primary uncertainty signal in GUI grounding, as formatting and descriptive tokens exhibit consistently high probabilities regardless of prediction quality. Figure 4 demonstrates this superiority persists across varying sampling budgets: C-Conf maintains its advantage from minimal sampling (N=2: 76.7% vs 74.8% for standard confidence) to larger ensembles (N=12: 83.9% vs 78.6%). Notably, while spatial methods like Center collapse at higher sampling rates due to averaging effects, C-Conf shows monotonic improvement, confirming that confidence-based selection scales effectively with computational budget. More experimental details can be found in the Appendix A.5.

**Distance threshold sensitivity reveals fundamental differences between positive and negative learning.** Table 3 examines how the threshold $\tau$ affects both methods across datasets. On ScreenSpot-V2, CRL exhibits strong sensitivity with performance degrading from 88.9% at $\tau = 0.05$ to 86.9% at $\tau = 0.1$, reflecting its reliance on accurate positive sample identification. Conversely, CNRL maintains remarkable stability at 88.5%±1.0% across all thresholds. This pattern intensifies on ScreenSpot-Pro where CRL's performance drops precipitously from 33.2% to 30.1% as $\tau$ increases, while CNRL remain stable. This invariance stems from a fundamental property of sparse coordinate spaces: predictions far from any reasonable target remain unambiguously incorrect regardless of the specific distance threshold. While positive samples require careful boundary definition to avoid including incorrect predictions, negative samples beyond any plausible threshold provide consistently reliable learning signals, making CNRL robust to hyperparameter selection.

Table 3: Distance threshold $\tau$ sensitivity analysis.

| Threshold $\tau$ | SS-V2 | SS-Pro |
|---|---|---|
| *CRL* | | |
| 0.01 | 87.7 | 33.3 |
| 0.03 | 88.4 | 33.2 |
| 0.05 | **88.9** | 33.2 |
| 0.07 | **88.9** | 30.1 |
| 0.1 | 86.9 | 30.7 |
| *CNRL* | | |
| 0.01 | 87.5 | 30.2 |
| 0.03 | 87.8 | 30.8 |
| 0.05 | 88.5 | **33.7** |
| 0.07 | 88.4 | 32.7 |
| 0.1 | 88.5 | 33.5 |

**Reward reliability validates the asymmetric quality of positive and negative samples.** Figure 5 quantifies the actual correctness of our binary reward assignments by comparing against ground truth labels. On ScreenSpot-V2 with larger bounding boxes, CNRL's reward accuracy falls slightly below CRL's because some correct predictions lying outside threshold $\tau$ are misclassified as negative. However, this apparent disadvantage transforms into a strength on ScreenSpot-Pro: CNRL maintains near-perfect negative sample reliability (>0.95) throughout training while CRL's mixed positive-negative accuracy hovers around 0.5. This empirical validation confirms our core hypothesis that high-resolution GUI tasks naturally provide accurate negative signals even without labels, as the vast coordinate space makes random distant predictions almost certainly incorrect.

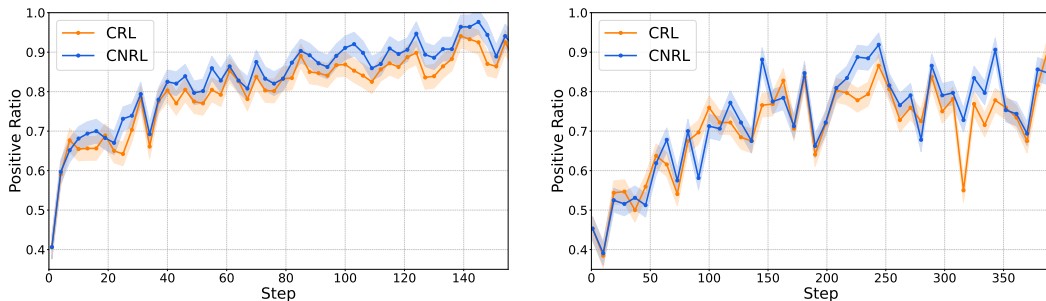

Figure 6: **Left**: Positive Ratio on ScreenSpot-V2. **Right**: Positive Ratio on ScreenSpot-Pro.

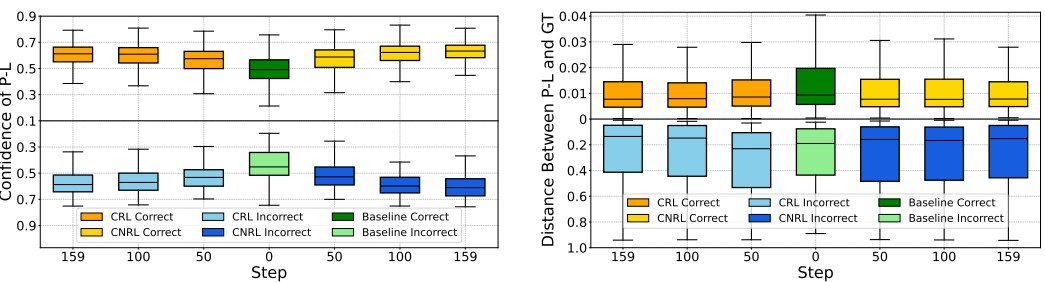

Figure 7: **Left**: Confidence of pseudo-label on ScreenSpot-V2. **Right**: Distance between the pseudo-label and the center of ground truth bounding box on ScreenSpot-V2.

**Training dynamics reveal distinct convergence patterns across difficulty levels.** Figure 6 tracks the evolution of positive ratio, defined as the fraction of samples within distance threshold $\tau$ from the pseudo-label. On ScreenSpot-V2, both methods exhibit monotonic increases, indicating progressive prediction concentration around confident regions. ScreenSpot-Pro presents markedly different dynamics: slower convergence with higher variance, plateauing around 0.8 rather than continuing upward. This divergence reflects the inherently greater challenge of high-resolution grounding where precise localization becomes critical. Notably, CNRL consistently maintains higher positive ratios than CRL across both settings, suggesting that learning exclusively from negative samples paradoxically helps models develop more concentrated predictions, possibly by more effectively pruning incorrect hypotheses from the prediction space.

**Pseudo-label characteristics reveal both successes and inherent limitations.** Figure 7 provides deeper insights into pseudo-label behavior. The left panel shows coordinate-token confidence increases for both correct and incorrect pseudo-labels during training, with correct predictions maintaining a consistent advantage. This convergence suggests models become increasingly confident regardless of accuracy, potentially limiting further improvements without external supervision. The right panel reveals a striking bimodal distribution: correct pseudo-labels cluster within 0.02 normalized distance of ground truth centers, while incorrect ones remain at 0.40 distance throughout training. This binary pattern validates our distance-based reward design and explains why negative samples are overwhelmingly reliable in sparse coordinate spaces. The persistence of distant incorrect pseudo-labels indicates certain challenging cases remain beyond the model's capability without ground truth, representing an inherent limitation of label-free learning.

**Cross-dataset experiments further demonstrate the generalization capability.** We train Qwen-2.5-VL-3B for 1 epoch on one dataset and then evaluate it on another. Figure 8 demonstrates the effectiveness of our approach under the cross-dataset setting. Owing to the high similarity between ScreenSpot-V1 and ScreenSpot-V2, as well as the presence of annotation errors in ScreenSpot-V1, we omit evaluation on ScreenSpot-V1. After training on UI-Vision, the performance improves by 5.3% on ScreenSpot-V2 (87.4% compared to 82.1%) and 15.6% on ScreenSpot-Pro (31.7% compared to 16.1%). Additionally, training on ScreenSpot-V2 results in a 4.6% performance improvement on UI-Vision (16.6% compared to 12.0%). The results indicate that our method is not merely overfitting the training dataset; rather, it exhibits strong generalization capability and achieves substantial improvements on previously unseen datasets. Moreover, when the training dataset is more

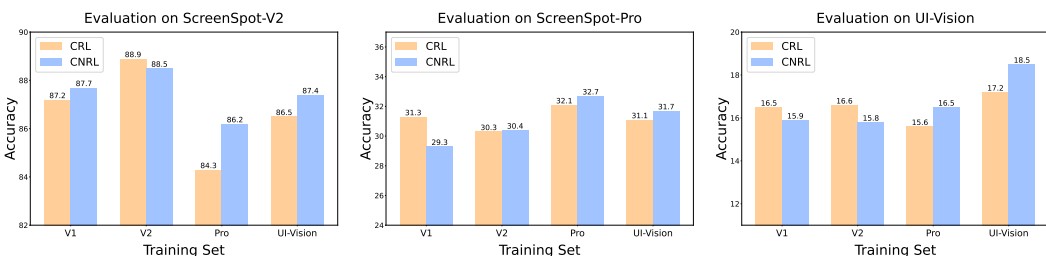

Figure 8: Cross-dataset evaluation.

challenging (ScreenSpor-Pro and UI-Vision), CNRL achieves better performance than CRL among evaluated benchmarks.

## 5 CONCLUSION

We presented a label-free training paradigm for GUI grounding that eliminates the dependency on costly pixel-level annotations. Our key innovation, coordinate-token confidence, provides a reliable signal for pseudo-label selection without ground truth. Building on this, we developed CRL for confidence-based reinforcement learning and CNRL for exclusive negative learning. Without any annotations, CNRL-7B achieves 92.1% on ScreenSpot-V2, surpassing UI-TARS-72B trained on 18.4M labels, and 33.8% on ScreenSpot-Pro, exceeding supervised baselines. The consistent superiority of CNRL on high-resolution benchmarks reveals that in sparse coordinate spaces, learning what to avoid provides more reliable supervision than learning from uncertain positive examples. Our analysis shows coordinate-token confidence outperforms alternative strategies by 2.1-11.9%, while CNRL's robustness to hyperparameters ensures practical deployment. These findings establish that internal model confidence can effectively replace manual annotations, opening new avenues for scaling GUI agents without the bottleneck of data annotation.

## 6 FUTURE WORK

This work presents a preliminary exploration of label-free learning for GUI grounding, demonstrating the feasibility of this training paradigm. Building on our work, we believe that extending this label-free training paradigm to large-scale training on public sets is a highly promising direction, as it can effectively reduce the reliance on manual annotations in GUI grounding tasks. On the other hand, extending this label-free training paradigm to end-to-end GUI Agents tasks in the future is also highly promising.

## ETHICS STATEMENT

This work focuses on advancing label-free reinforcement learning methods for GUI grounding tasks. Our research does not involve the collection or annotation of human subject data, nor does it utilize personally identifiable information. The datasets employed in this study are publicly available benchmarks, ensuring that no additional privacy or ethical risks are introduced. Potential misuse of our method, such as deploying GUI agents in malicious automation scenarios, should be carefully considered by practitioners. We encourage responsible application of our approach within research and development contexts that align with ethical guidelines and benefit broader society.

## REPRODUCIBILITY STATEMENT

To ensure reproducibility, we provide detailed descriptions of training configurations, hyperparameters, and evaluation protocols in the main paper and supplementary materials. All experiments are conducted on publicly available datasets. we report results averaged over multiple runs to account for variability. These steps are intended to facilitate faithful reproduction and fair comparison of our results.

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

# A   APPENDIX

## THE USE OF LARGE LANGUAGE MODELS (LLMS)

A LLM was employed exclusively for linguistic polishing of the manuscript, including improving grammar, fluency, and readability. The LLM did not contribute to research conception, methodology design, data analysis, or the generation of technical or scientific content. All substantive ideas, experiments, and conclusions presented in this paper are entirely the work of the authors. The use of the LLM was limited to editorial assistance to enhance the clarity of writing, in line with the policy on acceptable LLM usage.

## A.1   EVALUATION DETAILS

Following (Zuo et al., 2025), we independently apply our methods on each benchmark to implement test-time reinforcement learning. On each dataset, we use all available original inputs (image and instruction) on each benchmark for label-free training. This section provides an overview of the benchmarks.

- **ScreenSpot** (Cheng et al., 2024) is a widely used benchmark for GUI grounding, which contains 1272 instructions across mobile, desktop and web domains.
- **ScreenSpot-V2** (Wu et al., 2024b) is a enhanced version of ScreenSpot with error correction and re-annotation. It contains 1272 instructions across mobile, desktop and web domains.
- **ScreenSpot-Pro** (Li et al., 2025) is designed to rigorously evaluate the grounding capabilities of MLLMs in high-resolution professional settings. It contains 1581 samples, spanning 23 applications across five industries and three operating systems.
- **UI-Vision** (Nayak et al., 2025) is a comprehensive, license-permissive benchmark for offline, fine-grained evaluation of computer use agents in real-world desktop environments. It provides three fine-to-coarse grained tasks: (1) element grounding; (2) layout grounding; and (3) action prediction. Since our work focuses on the model's grounding capability for target elements, we use only the element grounding component during both training and evaluation. It introduce three grounding subtasks—basic, functional, and spatial—to assess different aspects of GUI understanding beyond simple textual queries. These three categories contain 1772, 1772, and 1935 instructions, respectively, totaling 5749 samples.

## A.2   SPARSE REWARDS VS. DENSE REWARDS UNDER LABEL-FREE SETTING

In our proposed methods, the pseudo label is represented as a single point, making reward assignment based on point-to-point distance a natural choice, which corresponds to a dense reward scheme. In addition, all points can be dilated into a region according to the distance threshold $\tau$, and the elliptical IoU between each region and the pseudo-label region can be computed as a dense reward. Dense rewards provide fine-grained supervisory signals, thereby facilitating more effective model learning. As shown in Figure 9, training with continuous rewards (Distance) accelerates convergence. However, as training progresses, its accuracy drops noticeably below that of binary rewards, as demonstrated in Table 4. Although the IoU-based reward scheme slows down the model's convergence, the resulting performance gains are noticeably smaller compared to binary reward. We contend that under noisy labels, finer reward granularity increases susceptibility to error. Specifically, continuous fine-grained rewards encourage the model to align outputs closely with the pseudo label. Even when the pseudo label lies within the target bounding box, it rarely coincides with the true center of the bounding box, thereby introducing bias. More critically, when the pseudo label falls outside the bounding box, continuous rewards exacerbate the issue by further misleading the model.

## A.3   PURE POSITIVE REINFORCEMENT LEARNING

In label-free settings, negative samples provide more reliable supervision than positive ones. To examine this core assumption, we conduct an ablation study employing pure positive reinforcement

Table 4: Accuracy of different reward type during training on ScreenSpot-V2.

| Reward Type | 50 Step | 100 Step | 159 Step |
|---|---|---|---|
| Binary Reward | 87.6 | 87.9 | 88.9 |
| Continuous Reward (Distance) | 87.7 | 86.3 | 85.2 |
| Continuous Reward (IoU) | 87.1 | 87.4 | 87.7 |

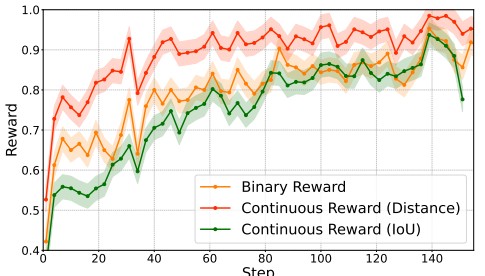 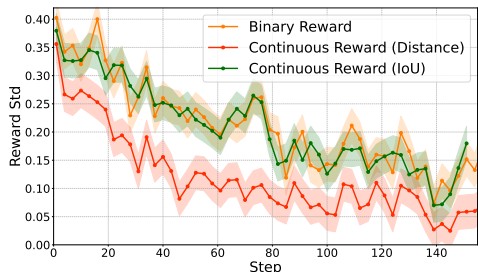

Figure 9: **Left**: Reward of binary reward and continuous reward during training on ScreenSpot-V2. **Right**: Reward std of binary reward and continuous reward during training on ScreenSpot-V2.

learning, which relies solely on pseudo-labeled positive samples. As illustrated in Figure 10, pure positive learning achieves faster convergence in terms of reward values. However, Table 5 demonstrates that the corresponding accuracy gains are limited and eventually decline as training proceeds. This discrepancy arises because rapid reward convergence reflects the model's tendency to concentrate predictions around potentially erroneous pseudo-labels, thereby overfitting to incorrect targets. Given the inevitable noise in pseudo-labels, exclusive reliance on positive samples misguides the learning process and leads to performance degradation over time. In contrast, CNRL converges slower, and performs significantly better than pure positive sample learning. These results validate our hypothesis that negative samples offer more robust learning signals in label-free scenarios, thereby justifying CNRL's design choice to focus exclusively on negative samples for policy optimization. Notably, due to the relative simplicity of ScreenSpot-V2, the positive samples also have a certain level of reliability. Training with both positive and negative samples, i.e. CRL—can further improve the model's performance.

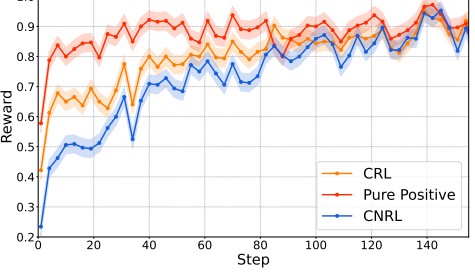 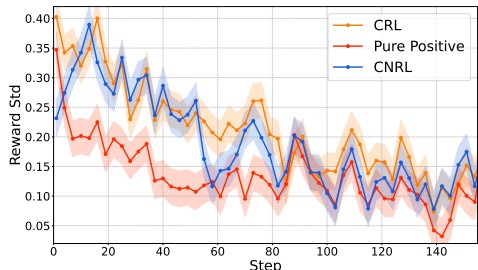

Figure 10: **Left**: Reward of CRL, CNRL and pure positive learning during training on ScreenSpot-V2. **Right**: Reward std of CRL, CNRL and pure positive learning during training on ScreenSpot-V2.

### A.4 REWARD ESTIMATION CASE STUDY

To understand the reliability of our distance-based reward assignment, we analyze three representative scenarios that commonly occur during pseudo-label generation:

Table 5: Accuracy of different sample selection mechanisms during training on ScreenSpot-V2.

| Method | 50 Step | 100 Step | 159 Step |
|---|---|---|---|
| Pure Positive | 86.3 | 87.2 | 86.7 |
| CRL | 87.6 | 87.9 | 88.9 |
| CNRL | 88.4 | 88.0 | 88.5 |

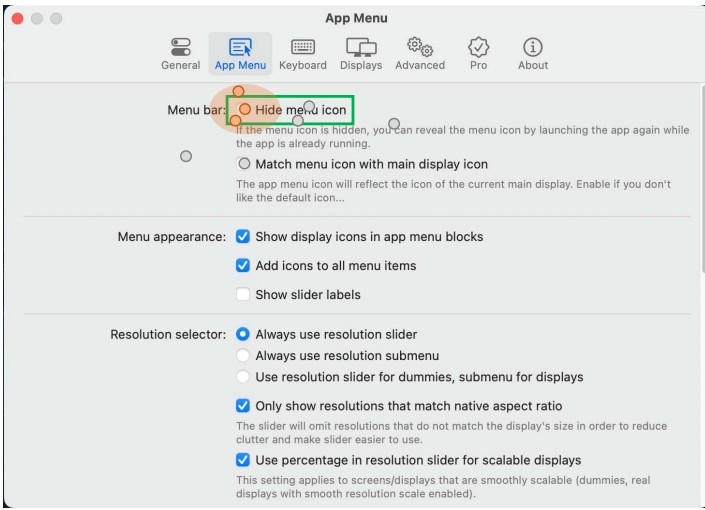

Figure 11: Reward estimation case 1. The green bounding box represents the ground truth, the orange region represents the positive sample area, the orange dots represent the samples evaluated as positive, and the gray dots represent the samples evaluated as negative.

- **Geometric mismatch (Case 1, Figure 11):** The pseudo-label falls within the target bounding box, but the elongated shape of the ground truth creates a mismatch where many samples that actually lie inside the rectangular target are classified as negative due to the circular reward region. This geometric misalignment leads to conservative reward assignment where correct predictions are penalized.
- **Good alignment (Case 2, Figure 12):** The pseudo-label and circular reward region align well with the target, resulting in accurate reward estimation where most positive samples correspond to correct predictions.
- **Misplaced pseudo-label (Case 3, Figure 13):** The pseudo-label is positioned far from the target bounding box, making all samples within the reward region incorrect, yet they receive positive rewards, demonstrating a failure case of pseudo-label generation.

These cases demonstrate the varying reliability of positive and negative samples under different pseudo-label quality conditions. The analysis reveals why CNRL's exclusive focus on negative samples provides more robust learning signals, particularly in Cases 1 and 3 where positive sample reliability is compromised.

### A.5 COMPARISON OF DIFFERENT PSEUDO-LABEL CONSTRUCTION METHOD

Table 6 and 7 shows the performance of different pseudo-label construction methods with different temperature and rollout number. Coordinate-token confidence, Confidence represents selecting by coordinate-token confidence and averaged all-token confidence respectively. Coordinate-token Entropy and Entropy indicates selecting by coordinate-token entropy and averaged all-token entropy respectively. Random means selecting a predicted point randomly. Majority Voting selects as the answer the point that has the largest number of neighboring points within a predefined distance threshold. Following (Lee et al., 2025), we applied Kde, Center and Medoid. Kde selects the point with highest estimated density in the plane. Center is simply getting the mean of all predicted co-

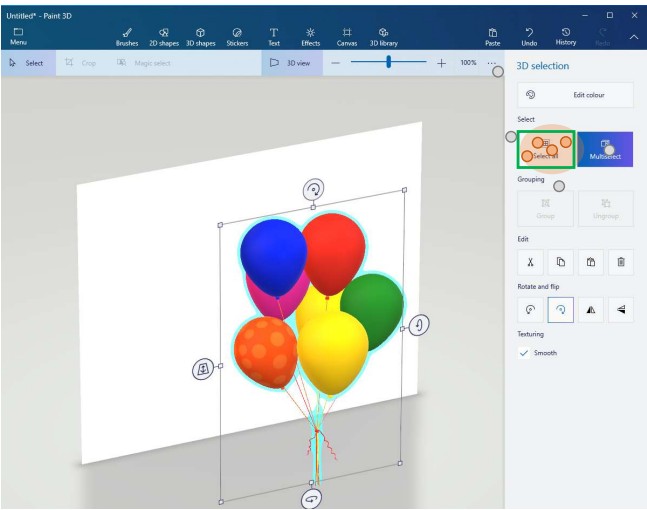

Figure 12: Reward estimation case 2. The green bounding box represents the ground truth, the orange region represents the positive sample area, the orange dots represent the samples evaluated as positive, and the gray dots represent the samples evaluated as negative.

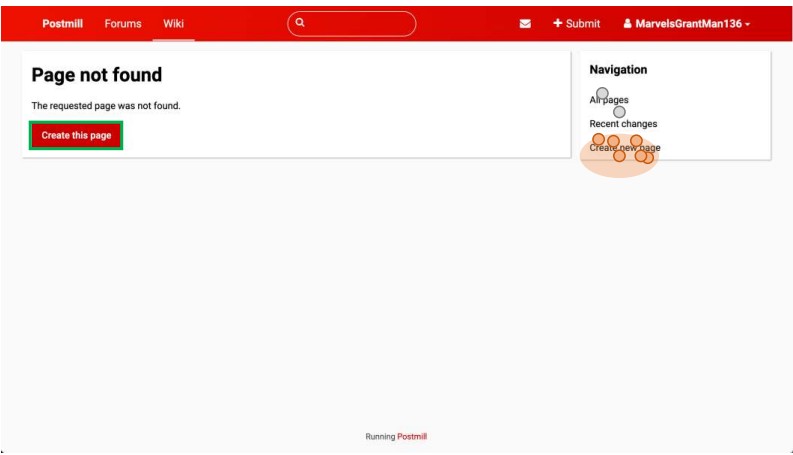

Figure 13: Reward estimation case 3. The green bounding box represents the ground truth, the orange region represents the positive sample area, the orange dots represent the samples evaluated as positive, and the gray dots represent the samples evaluated as negative.

Table 6: Performance of different pseudo-label construction methods with different temperature. Best and second-best results are shown in **Bold** and underline, respectively.

| Method | $T = 0.6$ | $T = 0.7$ | $T = 0.8$ | $T = 0.9$ | $T = 1.0$ |
|---|---|---|---|---|---|
| Coordinate-Token Confidence | **83.3** | **83.2** | **82.4** | **82.6** | **80.7** |
| Confidence | 82.1 | 80.3 | 77.7 | 74.2 | 71.7 |
| Coordinate-Token Entropy | 81.6 | 81.4 | 80.9 | 78.2 | 77.4 |
| Entropy | 82.1 | 81.6 | 80.3 | 77.7 | 76.6 |
| KDE | 82.2 | 82.1 | 79.7 | 80.0 | 78.2 |
| Center | 74.5 | 70.8 | 68.2 | 62.0 | 55.0 |
| Majority Voting | 81.6 | 80.5 | 80.0 | 80.3 | 79.6 |
| Medoid | 82.5 | 81.8 | 80.0 | 79.3 | 78.0 |
| Random | 79.0 | 75.5 | 73.3 | 69.7 | 67.0 |

Table 7: Performance of different pseudo-label construction methods with different rollout number $N$. Best and second-best results are shown in **Bold** and underline, respectively.

| Method | $N = 2$ | $N = 4$ | $N = 6$ | $N = 8$ | $N = 10$ | $N = 12$ |
|---|---|---|---|---|---|---|
| Coordinate-Token Confidence | **76.7** | **81.5** | **82.8** | **83.2** | **83.3** | **83.9** |
| Confidence | 74.8 | 78.4 | 80.0 | 80.3 | 78.9 | 78.6 |
| Coordinate-Token Entropy | 75.7 | 80.1 | 80.7 | 81.4 | 81.4 | 81.4 |
| Entropy | 75.6 | 79.9 | 80.5 | 81.6 | 82.0 | 81.4 |
| KDE | 75.6 | 80.1 | 81.4 | 82.1 | 81.3 | 81.1 |
| Center | 73.3 | 74.2 | 72.9 | 70.8 | 71.3 | 72.0 |
| Majority Voting | 72.7 | 79.0 | 80.7 | 80.5 | 81.5 | 80.9 |
| Medoid | 72.7 | 80.6 | 81.5 | 81.8 | 82.5 | 81.8 |
| Random | 73.5 | 74.5 | 76.5 | 75.5 | 76.4 | 75.9 |

ordinates. Medoid selects an actual predicted point that minimizes the sum of distances to all other predictions. As shown in Table 6, with the increase of temperature, the prediction points generated by the model become more inaccurate, leading to a decline in the performance of all methods. As the number of rollouts increases, the pseudo-labels constructed by each method become increasingly accurate, as shown in Table 7. Compared with other methods, our proposed approach achieves a substantial lead in performance under any temperature and number of rollouts, which demonstrated its effectiveness and robustness.

## A.6 Detailed Results of UI-Vision

UI-Vision represents the most challenging benchmark in our evaluation suite, featuring complex desktop applications and diverse interaction patterns across education, browser, development, productivity, creative, and entertainment categories. Table 9 presents detailed performance breakdowns across these categories and three difficulty settings (Basic, Functional, and Spatial). Our label-free methods demonstrate remarkable effectiveness on this challenging benchmark, with CNRL-7B achieving 20.1% overall accuracy compared to the base model's 15.0%, representing a 34% relative improvement without using any labeled data. The performance gains are particularly pronounced in categories requiring precise localization: Browser (38.5%), Entertainment (44.8%), and Productivity (20.9%). Notably, CNRL consistently outperforms CRL across almost all categories, with the advantage most significant in Entertainment tasks (44.8% vs 41.1%), validating that negative learning becomes

Table 8: Training hyperparameters.

| Hyperparameter | Value |
|---|---|
| $\tau$ | 0.05 |
| $\beta$ | 0.04 |
| $T$ | 1.0 |
| $top\_k$ | 50 |
| $top\_p$ | 1.0 |
| learning_rate | 1e-6 |
| bf16 | true |
| torch_dtype | bfloat16 |
| data_seed | 42 |
| gradient_checkpointing | true |
| attn_implementation | flash_attention_2 |
| num_train_epochs | 1 |
| max_pixels | 12845056 |

Table 9: Performance comparison on UI-Vision.

| Model | Labels | Grouped by Category | | | | | | Grouped by Setting | | | Overall |
|---|---|---|---|---|---|---|---|---|---|---|---|
| | | Edu. | Browser | Dev. | Prod. | Creative | Entert. | Basic | Func. | Spatial | |
| *Proprietary Models* | | | | | | | | | | | |
| GPT-4o | - | 1.5 | 0.0 | 2.2 | 1.1 | 0.8 | 4.2 | 1.6 | 1.5 | 1.0 | 1.4 |
| Claude Computer Use | - | 6.1 | 9.8 | 8.0 | 9.4 | 7.7 | 8.3 | 9.5 | 7.7 | 7.6 | 8.3 |
| *Models Trained on Labeled GUI Data* | | | | | | | | | | | |
| SeeClick-9.6B | 1M | 4.2 | 13.3 | 7.3 | 4.3 | 4.0 | 11.0 | 9.4 | 4.7 | 2.1 | 5.4 |
| ShowUI-2B | 256K | 3.7 | 13.3 | 7.5 | 6.5 | 2.5 | 15.6 | 8.1 | 7.7 | 2.1 | 5.9 |
| CogAgent-9B | 222M | 8.7 | 11.2 | 8.6 | 10.3 | 5.6 | 15.6 | 12.0 | 12.2 | 2.6 | 8.9 |
| OSAtlas-7B | 13M | 8.7 | 16.8 | 10.3 | 9.2 | 5.6 | 16.2 | 12.2 | 11.2 | 3.7 | 9.0 |
| AriaUI | 17.6M | 9.0 | 18.9 | 11.2 | 10.4 | 6.5 | 19.3 | 12.2 | 14.0 | 4.0 | 10.1 |
| UGround-v1-7B | - | 10.4 | 28.7 | 17.5 | 12.2 | 8.6 | 18.2 | 15.4 | 17.1 | 6.3 | 12.9 |
| Aguvis-7B | 1M | 13.1 | 30.8 | 17.1 | 12.1 | 9.6 | 24.0 | 17.8 | 18.3 | 5.1 | 13.7 |
| UI-TARS-7B | 18.4M | **14.2** | **35.0** | 19.7 | 18.3 | **11.1** | 38.5 | 20.1 | 24.3 | 8.4 | **17.6** |
| *Vanilla Models* | | | | | | | | | | | |
| Qwen-2.5-VL-3B | 0 | 7.6 | 22.4 | 15.4 | 12.8 | 6.6 | 33.9 | 18.6 | 13.8 | 4.3 | 12.0 |
| Qwen-2.5-VL-7B | 0 | 11.1 | 37.1 | 18.1 | 15.4 | 9.6 | 29.7 | 20.0 | 18.6 | 7.1 | 15.0 |
| *Ours* | | | | | | | | | | | |
| CRL-3B | 0 | 15.1 | 32.9 | 20.6 | 18.1 | 10.4 | 37.0 | 24.9 | 22.1 | 5.7 | 17.2 |
| CNRL-3B | 0 | 16.0 | 33.6 | 21.6 | 19.4 | 12.0 | 39.1 | 26.4 | 23.8 | 6.5 | 18.5 |
| CRL-7B | 0 | **17.6** | **42.7** | 22.0 | 18.9 | 10.9 | 41.1 | 25.6 | 22.9 | 8.4 | 18.6 |
| CNRL-7B | 0 | 17.3 | 38.5 | **24.1** | **20.9** | **12.4** | **44.8** | **27.5** | **25.1** | **8.8** | **20.1** |

increasingly beneficial as coordinate spaces expand and targets become more sparse. The superior performance of CNRL over supervised baselines like ShowUI-2B (5.9%) and its competitive results against models trained on millions of labels demonstrates that confidence-guided negative reinforcement learning provides a viable path for label-free GUI grounding even in the most demanding scenarios.

## A.7    REINFORCEMENT LEARNING VS. SUPERVISED FINE TUNING

We construct pseudo-labels for Supervised Fine-Tuning (SFT) using the same sampling parameters as those in RL (e.g., temperature, rollout), and adopt identical training parameters (e.g., learning rate). The results are shown in Table 10. Experimental results show that SFT (84.6%) performs significantly worse than RL (88.5%, 88.9%), indicating that in pseudo-label settings, RL is more effective in eliciting the model's potential. Unlike the absolute targets of SFT, GRPO optimizes the policy using relative advantage. This relative, normalized optimization naturally dampens the undue influence of individual noisy pseudo-labels, allowing the RL approaches to achieve superior performance in the presence of imperfect self-generated data.

Table 10: Accuracy of RL and SFT on ScreenSpot-V2.

| Method | Accuracy |
|---|---|
| Qwen-2.5-VL-3B | 82.1 |
| + CRL | **88.9** |
| + CNRL | 88.5 |
| + SFT | 84.6 |

## A.8    COMPARISON OF DIFFERENT COORDINATE-TOKEN CONFIDENCE CALCULATION

Prior work has proposed multiple approaches for computing confidence scores. Given the probability values of the generated coordinate tokens, we evaluate three confidence estimation methods: (1) Mean, which is the approach adopted in our training; (2) Product, which uses the N-th root of

Table 11: Accuracy of different confidence calculation methods on ScreenSpot-V2.

| Method | Accuracy |
|---|---|
| Mean | **80.7** |
| Product | 80.4 |
| Max | 76.4 |
| Min | 78.1 |

product of all coordinate-token probabilities as the confidence; (3) Max, which selects the highest probability among all coordinate-token probabilities; and (4) Min, which selects the lowest probability among all coordinate-token probabilities. We conduct experiments with temperature $T = 1.0$ and rollout number $N = 8$, and the results are presented in Table 11. Experimental results indicate that directly computing the mean probability yields the most effective confidence measure. It strikes the best balance between penalizing uncertainty, providing the most reliable signal for pseudo-label selection.

## A.9 TRAINING ON PUBLIC DATASET

To further investigate the generalization capability of the proposed method, we conducted training on GroundCUA (Feizi et al., 2025), which is a publicly available datasets. We randomly selected 5k samples from the dataset for training, and the results are shown in the Table 12. The results show that our method exhibits strong generalization capability; even when trained on publicly available datasets, it achieves substantial performance gains.

Table 12: Evaluation of our method trained on publicly available datasets.

| Method | ScreenSpot | ScreenSpot-V2 | ScreenSpot-Pro | UI-Vision |
|---|---|---|---|---|
| Qwen-2.5-VL-3B | 77.6 | 82.1 | 16.1 | 12.0 |
| + CRL | **84.6** | **87.3** | 32.6 | **16.7** |
| + CNRL | 83.5 | 86.7 | **33.4** | **16.7** |

## A.10 DOWNSAMPLING STRATEGY

During training, we sample 16 responses to construct pseudo-labels, followed by downsampling 8 to optimization, which is similar to Zuo et al. (2025). It is a critical mechanism for overcoming the fundamental challenge of no ground-truth data in our label-free setting. Sampling 16 responses (rather than 8) is crucial for two reasons: it generates a sufficient variety of predictions, which is essential for discovering high-quality pseudo-labels and effective negative samples. Furthermore, downsampling 8 responses for optimization improves computational efficiency (in contrast to optimizing with all 16 responses) and enhances the negative signal quality for CNRL. These 8 samples are not random, but are specifically selected as the farthest from the chosen pseudo-label. By restricting the policy update to these most distant samples, the negative reward signals are derived from the highest quality "incorrect" predictions, making the penalty mechanism highly effective for correcting errors.

## A.11 TRAINING HYPERPARAMETERS

We provide detailed hyperparameter configurations to ensure reproducibility of our label-free training approach. Table 8 presents the core training parameters used across all experiments. The distance threshold $\tau$=0.05 was selected based on preliminary experiments to balance between capturing genuine positive samples and maintaining negative sample reliability. We employ a relatively conservative learning rate of 1e-6 with KL penalty $\beta$=0.04 to ensure stable policy updates during reinforcement learning. Sampling parameters (T=1.0, top_k=50, top_p=1.0) are configured to maximize response diversity, which proves critical for generating varied candidates for pseudo-label selection. To accommodate different dataset characteristics, we adjust gradient accumulation steps:

8 for ScreenSpot-V2, 4 for ScreenSpot-Pro, and 16 for UI-Vision, ensuring consistent effective batch sizes despite varying computational demands. All experiments utilize bfloat16 mixed precision training with gradient checkpointing and Flash Attention 2 for memory efficiency, enabling training on consumer-grade GPUs. The prompts are tailored to model capacity, with 3B models using a simpler single-point format while 7B models support multi-element detection, though both maintain the same coordinate-based output structure essential for our confidence computation.

---

**3B Model Prompt**

```
point to the instruction:  {Question}, output its coordinates
in JSON format {{"point_2d":  [x, y], "label":  "object
name/description"}}.
```

---

**7B Model Prompt**

```
Locate the UI element(s) for {Question}, output the
coordinates using JSON format:  [{{"point_2d":  [x, y]}}, ...]
```