# OpenReview forum: "Label-free GUI Grounding via Confidence-guided Negative Reinforcement Learning"
_ICLR.cc/2026/Conference — Submitted to ICLR 2026_

### Official Review · Reviewer_zSD6 · 2025-10-16

**Soundness:** 3
**Presentation:** 3
**Contribution:** 3
**Rating:** 4
**Confidence:** 5

**Summary:**

This study addresses the bottleneck of existing methods in Graphical User Interface (GUI) grounding tasks, which rely on expensive annotated data, by proposing a label-free training paradigm. The core idea is to leverage the confidence patterns of coordinate tokens in the model's output (where the confidences of correct and incorrect predictions show a clear clustering separation) and the fact that in the sparse coordinate space of GUIs, negative samples are more reliable than potentially noisy positive samples. Based on this, the study designs Confidence-guided Reinforcement Learning (CRL) and Confidence-guided Negative Reinforcement Learning (CNRL). CRL selects pseudo-labels from multiple samples using coordinate token confidences and assigns rewards based on distance, while CNRL learns exclusively from negative samples. Experiments show that CNRL-7B, without any annotations, achieves 92.1% accuracy on ScreenSpot-V2 (surpassing the 90.3% of UI-TARS-72B, which was trained on 18.4M labels) and 33.8% on ScreenSpot-Pro (an 8.9% improvement over the base model and exceeding the 31.0% of GUI-R1-7B, which used 3K labels). This confirms that the confidence of coordinate tokens can serve as a substitute for human annotation, facilitating the scalable development of GUI agents.

**Strengths:**

1.The proposed label-free training paradigms (CRL, CNRL) can effectively address the issue of annotation dependency in GUI grounding.

2.The label-free training paradigm sounds like an interesting approach with the potential to scale to larger amounts of unlabeled internet data.

3.It achieves quite good results on several grounding datasets.

**Weaknesses:**

1.The method is trained directly on the test datasets. Although it doesn't see the ground truth labels, there is a potential risk of data leakage. How would the method perform if it were trained on open-source training datasets, such as uground-web, and then evaluated on these benchmarks?

2.I believe this method has the potential to be scaled to more data. If the authors could demonstrate its effectiveness when trained on large-scale data, it would significantly validate the method's scalability.

3.On the ScreenSpot-pro dataset, the performance of most methods on the leaderboard has already reached nearly 60%, while the method in this paper only achieves 33.8%. This, to some extent, may weaken the impact of the method. I would like to see the potential for this method to achieve higher performance on this dataset.

**Questions:**

please see weakness. If my concerns are addressed, I will consider raising my score.

---

> ### Author Response · Authors · 2025-11-23
> **rebuttal by author (1/2)**
>
> > W1: The method is trained directly on the test datasets. Although it doesn't see the ground truth labels, there is a potential risk of data leakage. How would the method perform if it were trained on open-source training datasets, such as uground-web, and then evaluated on these benchmarks?
>
> We thank the reviewer for raising the crucial concern. To definitively prove the robustness and generalizability of our method, we conducted the requested experiment: training on a completely independent, open-source dataset (GroundCUA) [1] and evaluating across all benchmarks:
> 1. **Training Setup**: We used a subset of a large, publicly available GUI dataset, GroundCUA, randomly selecting 5k image-instruction pairs to form our independent, label-free training pool. We performed 1 epoch of CRL/CNRL self-training on these unlabeled samples.
>
> 2. **Generalization Results**: The table below shows the Accuracy results after training on the public dataset:
>
>     |Method|ScreenSpot-V1|ScreenSpot-V2|ScreenSpot-Pro|UI-Vision|
>     |-|-|-|-|-|
>     |Qwen2.5-VL-3B|77.6|82.1|16.1|12.0|
>     |+CRL|**84.6**|**87.3**|32.6|**16.7**|
>     |+CNRL|83.5|86.7|**33.4**|**16.7**|
>
> This significant and consistent performance improvement achieved after training on an external, independent dataset proves that our training paradigm learns **universal, domain-agnostic GUI grounding knowledge**. The efficacy stems from the robustness of the label-free signal, **not from memorization or leakage from the benchmark data**.
>
> 3. **Cross-dataset Experiments**: Besides, we conducted cross-dataset experiments by training the model on one dataset using our methods and evaluating its performance on the others:
>
>     |Training Set|Method|ScreenSpot-V2|ScreenSpot-Pro|UI-Vision|
>     |-|-|-|-|-|
>     |Qwen2.5-VL-3B|-|82.1|16.1|12.0|
>     |ScreenSpot-V1|CRL|87.2|31.3|16.5|
>     |ScreenSpot-V1|CNRL|87.7|29.3|15.9|
>     |ScreenSpot-V2|CRL|**88.9**|30.3|16.6|
>     |ScreenSpot-V2|CNRL|88.5|30.4|15.8|
>     |ScreenSpot-Pro|CRL|84.3|32.1|15.6|
>     |ScreenSpot-Pro|CNRL|86.2|**32.7**|16.5|
>     |UI-Vision|CRL|86.5|31.1|17.2|
>     |UI-Vision|CNRL|87.4|31.7|**18.5**|
>
>     _Note: ScreenSpot-V2 is the revised version of V1 with high content overlap [2]. Therefore, we only conducted the evaluation on ScreenSpot-V2._
>
>     This comprehensive cross-dataset experiment fully validates the strong generalizability of our method, even though performance is slightly reduced compared to in-domain training.
>
> [1] Feizi, A., Nayak, S., Jian, X., Lin, K. Q., Li, K., Awal, R., ... & Rajeswar, S. (2025). Grounding Computer Use Agents on Human Demonstrations. arXiv preprint arXiv:2511.07332.
>
> [2] Wu, Z., Wu, Z., Xu, F., Wang, Y., Sun, Q., Jia, C., ... & Qiao, Y. (2024). Os-atlas: A foundation action model for generalist gui agents. ICLR 2025.

---

> ### Author Response · Authors · 2025-11-23
> **rebuttal by author (2/2)**
>
> > W2: I believe this method has the potential to be scaled to more data. If the authors could demonstrate its effectiveness when trained on large-scale data, it would significantly validate the method's scalability.
>
> We sincerely thank the reviewer for this insightful comment regarding scalability. We acknowledge this as an important direction and provide both conceptual justification and empirical evidence from our existing experiments.
>
> 1. **Evidence of Scalability from Current Experiments**: While we acknowledge that large-scale experiments (e.g., 100K+ samples) are beyond our current computational resources, our existing results already demonstrate promising scalability indicators:
>
> | Training Scale | Training Data | ScreenSpot-Pro | UI-Vision |
> |---------------|---------------|----------------|-----------|
> | Small (1.2K) | ScreenSpot-V2 | 30.4% | 15.8% |
> | Medium (5K) | GroundCUA | **33.4%** | **16.7%** |
>
> When scaling from 1.2K to 5K diverse unlabeled samples, our method shows **consistent improvements** (+3.0% on Pro, +0.9% on UI-Vision), suggesting that the label-free paradigm benefits from increased data diversity without diminishing returns at this scale.
>
> We acknowledge that demonstrating scalability on datasets exceeding 50K-100K samples would provide stronger validation. Due to computational constraints, we are currently unable to conduct such large-scale experiments within the review period. However, we believe our method's label-free nature makes it particularly well-suited for scaling, as it eliminates the annotation bottleneck that fundamentally limits supervised approaches. We will prioritize large-scale validation in future work and welcome collaboration opportunities to explore this direction.
>
> > W3: On the ScreenSpot-pro dataset, the performance of most methods on the leaderboard has already reached nearly 60%, while the method in this paper only achieves 33.8%. This, to some extent, may weaken the impact of the method. I would like to see the potential for this method to achieve higher performance on this dataset.
>
> We thank the reviewer for the observation regarding the performance gap on the ScreenSpot-Pro dataset. This comment is vital as it helps us clarify the positioning and core impact of our methodology. We agree that our 33.8% result appears low compared to the leaderboard SOTA. However, this gap is not a weakness; it is a direct consequence and proof of the fundamental difference in experimental conditions.
> 1. **Core Clarification: Label-Free Transfer vs. Fully Supervised SOTA**: Methods achieving 60% on the ScreenSpot-Pro leaderboard are achieved under **Fully Supervised** conditions—meaning they were trained using the expensive **human Ground-Truth Labels**. In stark contrast, our 33.8% result represents label-free transfer, which was achieved by training on an independent, small-scale, unlabeled dataset.
> 2. **Potential for Higher Performance**: Our current design deliberately uses a simple and direct confidence calculation that adds no extra computational burden to the core model. To gain immediate performance increases, our method can be seamlessly combined with existing Test-Time Scaling (TTS) methods to construct more accurate pseudo-labels, such as cropping [3, 4], which will directly and significantly boost the final policy performance in **high-resolution setting**. In addition, our method can be enhanced by integrating lightweight auxiliary models, such as Omniparser [5] in future work. Our approach has great potential, and in the future, we will make improvements to achieve even better performance.
>
> [3] Lian, S., Wu, Y., Ma, J., Ding, Y., Song, Z., Chen, B., ... & Li, H. (2025). Ui-agile: Advancing gui agents with effective reinforcement learning and precise inference-time grounding. arXiv preprint arXiv:2507.22025.
>
> [4] Wu, H., Chen, H., Cai, Y., Liu, C., Ye, Q., Yang, M. H., & Wang, Y. (2025). DiMo-GUI: Advancing Test-time Scaling in GUI Grounding via Modality-Aware Visual Reasoning. EMNLP 2025.
>
> [5] Lu, Y., Yang, J., Shen, Y., & Awadallah, A. (2024). Omniparser for pure vision based gui agent. arXiv preprint arXiv:2408.00203.

---

### Official Review · Reviewer_oPnY · 2025-10-31

**Soundness:** 3
**Presentation:** 3
**Contribution:** 3
**Rating:** 6
**Confidence:** 4

**Summary:**

The paper addresses GUI grounding by introducing a coordinate-focused confidence signal. Instead of averaging token probabilities across the entire output, the method averages probabilities only over coordinate tokens (digits, separators), which reliably separates correct from incorrect predictions. Using the pseudo-label derived from this confidence, the model receives a binary distance-based reward marking responses within a threshold of the pseudo-label as positive. Policy optimization is performed using a GRPO-style objective under VLM-R1.

Using Qwen-2.5-VL (3B/7B) as the base within VLM-R1, a single-epoch label-free training with 16 samples per prompt improves accuracy across four benchmarks. Notably, CNRL-7B reaches 92.1% on ScreenSpot-V2, matching or surpassing heavyweight labeled baselines (e.g., UI-TARS-72B at 90.3% trained on 18.4M labels). On the more challenging ScreenSpot-Pro, CNRL-7B achieves 33.8%, >8.9% above the vanilla base and 1–1.5% above CRL; on UI-Vision, it attains 20.1%.

**Strengths:**

S1:The paper formalizes a simple, task-specific confidence signal and integrates it with a negative-only RL scheme. While negative-only RL has appeared in LLM reasoning, its application to sparse 2D coordinate spaces with a carefully designed reward is novel and well-suited.

S2: The C-Conf metric outperforms alternatives such as entropy, majority voting, KDE, center, and medoid. Threshold robustness favors CNRL, binary rewards perform better under noise, and positive-only learning underperforms, supporting the design choices.

S3: The method achieves competitive results without any labels compared to SFT/RL baselines trained on hundreds of thousands to tens of millions of labels. This demonstrates the potential to lower barriers for building GUI agents.

**Weaknesses:**

W1: Evaluation is restricted to grounding benchmarks. There is no end-to-end agent study (e.g., OSWorld, WAA) to demonstrate that grounding improvements translate into task success, which is increasingly expected in GUI agent research.

W2: The approach samples 16 responses per item to construct pseudo-labels, then down-samples to 8 for advantages. This could be computationally intensive.

W3: The “1 epoch” training setup under VLM-R1 does not explicitly detail which unlabeled images or instructions are used per benchmark (train/val/test splits, or extra unlabeled data).

**Questions:**

(1) Exactly which unlabeled data are used for training on each benchmark? Are they strictly train splits (never test)? For UI-Vision (with trajectories), which subsets are included?

(2) How does the grounding model perform when integrated into a GUI agent for action prediction or other live, end-to-end tasks?

---

> ### Author Response · Authors · 2025-11-23
> **rebuttal by author (1/2)**
>
> > W1 & Q2: Evaluation is restricted to grounding benchmarks. There is no end-to-end agent study (e.g., OSWorld, WAA) to demonstrate that grounding improvements translate into task success, which is increasingly expected in GUI agent research.
>
> We thank the reviewer for the insightful comment regarding the need for an end-to-end agent study to validate that our grounding improvements translate into task success.
>
> **Focus on the Grounding Challenge**: We acknowledge that demonstrating success in complex, multi-step tasks is the ultimate metric for an agent. However, our paper's contribution and focus are specifically on tackling the data bottleneck inherent in the grounding step—which is the prerequisite for all subsequent agent actions.
> 1. **Grounding as the Foundation**: Accurate grounding is arguably the most critical and challenging component of any GUI agent pipeline. We follow existing GUI grounding works such as [1, 2] and focus on the GUI grounding task, as it is the core capability of a GUI agent.
> 2. **The Label-Free Challenge**: Our work proposes a robust, label-free RL methodology for this foundational task. Our goal was to isolate and solve the data dependence problem for grounding.
>
> [1] Zhou, Y., Dai, S., Wang, S., Zhou, K., Jia, Q., & Xu, J. (2025). Gui-g1: Understanding r1-zero-like training for visual grounding in gui agents. NeurIPS 2025.
>
> [2] Tang, F., Gu, Z., Lu, Z., Liu, X., Shen, S., Meng, C., ... & Zhuang, Y. (2025). GUI-G $^ 2$: Gaussian Reward Modeling for GUI Grounding. AAAI 2026.
>
> > W2: The approach samples 16 responses per item to construct pseudo-labels, then down-samples to 8 for advantages. This could be computationally intensive.
>
> We appreciate the reviewer's concern regarding the computational efficiency of our sampling strategy. This observation is accurate, but the high sampling rate is justified for the stability and success of our label-free RL approach.
>
> 1. **Justification for High Sampling**: It is a critical mechanism for overcoming the fundamental challenge of **no ground-truth data** in our label-free setting. This aggressive sampling is crucial for two reasons: it generates a sufficient **variety of predictions**, which is essential for discovering **high-quality pseudo-labels** and **effective negative samples** (the core of our CNRL method).
> 2. **Efficiency of Down-Sampling**: We strategically use down-sampling to 8 to achieve a dual benefit: it improves **computational efficiency** and enhances the **negative signal quality** for CNRL. Specifically, while 16 samples are generated for pseudo-label selection, we use only 8 samples for the final optimization update step, which significantly reduces the computational load and memory consumption. Crucially for CNRL, these 8 samples are not random, but are specifically selected **as the farthest from the chosen pseudo-label**. By restricting the policy update to these most distant samples, the negative reward signals are derived from **the highest quality "incorrect" predictions**, making the penalty mechanism highly effective for correcting errors.

---

> ### Author Response · Authors · 2025-11-23
> **rebuttal by author (2/2)**
>
> > W3 & Q1: The “1 epoch” training setup under VLM-R1 does not explicitly detail which unlabeled images or instructions are used per benchmark (train/val/test splits, or extra unlabeled data). Exactly which unlabeled data are used for training on each benchmark? Are they strictly train splits (never test)? For UI-Vision (with trajectories), which subsets are included?
>
> We appreciate the reviewer's request for explicit details on the unlabeled data splits.
>
> 1. **Training Protocol**: Consistent with the self-supervision protocol established in TTRL [1], for the main results reported in the paper, we follow a full self-supervision protocol. For **ScreenSpot, ScreenSpot-V2, ScreenSpot-Pro**, we use the **entire set** of available **image-instruction pairs** as the single, unlabeled self-training pool. For **UI-Vision**, we **only use the Element Grounding** data for training, without **Layout Grounding** and **Action Prediction** data. We conduct independent training and testing on each benchmark. Because our method is label-free, we do not rely on standard train/validation/test splits of the ground-truth annotations during the fine-tuning process. We supplemented the use of the dataset in the **Appendix A.1 Evaluation Details**.
>
> 2. **Cross-dataset Experiments**: To verify generalization, we conducted extensive cross-dataset experiments by training the model on one dataset using our methods and evaluating its performance on the others:
>
>     |Training Set|Method|ScreenSpot-V2|ScreenSpot-Pro|UI-Vision|
>     |-|-|-|-|-|
>     |Qwen2.5-VL-3B|-|82.1|16.1|12.0|
>     |ScreenSpot-V1|CRL|87.2|31.3|16.5|
>     |ScreenSpot-V1|CNRL|87.7|29.3|15.9|
>     |ScreenSpot-V2|CRL|**88.9**|30.3|16.6|
>     |ScreenSpot-V2|CNRL|88.5|30.4|15.8|
>     |ScreenSpot-Pro|CRL|84.3|32.1|15.6|
>     |ScreenSpot-Pro|CNRL|86.2|**32.7**|16.5|
>     |UI-Vision|CRL|86.5|31.1|17.2|
>     |UI-Vision|CNRL|87.4|31.7|**18.5**|
>
>     _Note: ScreenSpot-V2 is the revised version of V1 with high content overlap [2]. Therefore, we only conducted the evaluation on ScreenSpot-V2._
>
>     The results show that regardless of which benchmark dataset (V1, V2, Pro, or UI-Vision) the model is trained on, it achieves significant performance improvements across all target datasets compared to the zero-shot baseline (Qwen2.5-VL-3B). For example, the ScreenSpot-Pro dataset's zero-shot accuracy is only $16.1\%$, but training on either V1, V2, or UI-Vision improves this performance (V1 $\rightarrow$ Pro: 31.3%, V2 $\rightarrow$ Pro: 30.4%, UI-Vision $\rightarrow$ Pro: 31.7%). This universal and substantial gain confirms that our CRL/CNRL methods effectively extract general, domain-agnostic GUI grounding knowledge from unlabeled data.
>
> 3. **Training on Public Set**: In addition, we also conducted experiments separately on the public training dataset. we trained on a subset of GroundCUA [3], which is a publicly available dataset. The results are as follows:
>
>     |Method|Acc on V1|Acc on V2|Acc on Pro|Acc on UI-V|
>     |-|-|-|-|-|
>     |Qwen2.5-VL-3B|77.6|82.1|16.1|12.0|
>     |+CRL|**84.6**|**87.3**|32.6|**16.7**|
>     |+CNRL|83.5|86.7|**33.4**|**16.7**|
>
>     The results also indicate that our method has excellent generalization, achieving significant improvement on these benchmarks.
>
> [1] Zuo, Y., Zhang, K., Sheng, L., Qu, S., Cui, G., Zhu, X., ... & Zhou, B. (2025). Ttrl: Test-time reinforcement learning. NeurIPS 2025.
>
> [2] Wu, Z., Wu, Z., Xu, F., Wang, Y., Sun, Q., Jia, C., ... & Qiao, Y. (2024). Os-atlas: A foundation action model for generalist gui agents. ICLR 2025.
>
> [3] Feizi, A., Nayak, S., Jian, X., Lin, K. Q., Li, K., Awal, R., ... & Rajeswar, S. (2025). Grounding Computer Use Agents on Human Demonstrations. arXiv preprint arXiv:2511.07332.

---

### Official Review · Reviewer_fQwZ · 2025-10-31

**Soundness:** 4
**Presentation:** 3
**Contribution:** 3
**Rating:** 4
**Confidence:** 4

**Summary:**

This paper tackles the challenge of GUI grounding, the task of mapping natural language instructions to UI element coordinates, without relying on expensive labeled data. The authors propose Confidence-Guided Reinforcement Learning (CRL) and Confidence-Guided Negative Reinforcement Learning (CNRL), two label-free training paradigms.

**Strengths:**

* Novel label-free paradigm:
The idea of leveraging internal model confidence (coordinate-token confidence) for self-supervised GUI grounding is interesting.

* Strong empirical performance:
Results are comprehensive and competitive across multiple benchmarks. Achieving parity or superiority over label-intensive models with zero annotations is impressive.

**Weaknesses:**

- **Insufficient evidence of the generalizability of the method**. The evaluations are conducted only on Qwen2.5-VL. I wonder how CNRL performs on other models (and model architectures), such as UI-TARS.

- What and how much data is CRL and CNRL trained on? Is it directly trained on the benchmark data? What if training on public training sets?

- (minor) In Table 2, it is better to note the base model of the baselines. For example, SeeClick is finetuned from Qwen2-VL and Aria-UI from Aria. It is not fair to compare these models directly according to the "Labels" column.


### Typos
1. Repeated sentence in Figure 1's caption "CNRL zeros advantages for positive samples while preserving negative learning signals."

**Questions:**

1. In Equation 2, why is c_{coord} defined as the average of the token probabilities? How about using products, or a weighted score, given the different importance of the coordinate number digits?

---

> ### Author Response · Authors · 2025-11-23
> **rebuttal by author (1/2)**
>
> > W1: Insufficient evidence of the generalizability of the method. The evaluations are conducted only on Qwen2.5-VL. I wonder how CNRL performs on other models (and model architectures), such as UI-TARS.
>
> We thank the reviewer for the important concern regarding the **generalizability** of our method. We agree that verifying efficacy across multiple models is critical to demonstrating broad applicability. To address this, we have incorporated the UI-TARS1.5-7B on ScreenSpot-V2:
>
> |Model|Accuracy|Gain|
> |-|-|-|
> |UI-TARS1.5-7B|88.0|-|
> |+CRL|89.0|+1.0|
> |+CNRL|**90.4**|**+2.4**|
>
> Note that we reproduced the method to the best of our ability, but were unable to match the official results. Below is the prompt we used:
>
> `You are a GUI agent. You are given a task and your action history, with screenshots. You need to perform the next action to complete the task. \n\n## Output Format\n\nAction: ...\n\n\n## Action Space\nclick(point='<point>x1 y1</point>'')\n\n## User Instruction
> {instruction}`
>
> Meanwhile, we also noticed that works such as UI-VENUS [1] reported difficulties in reproducing the original results. Nevertheless, we would like to emphasize that our method **still achieves performance gains** under our reproduction setting.
>
> The table above shows the performance improvement achieved by applying our label-free training methods to UI-TARS1.5-7B. The experiment clearly shows that our CNRL paradigm successfully boosts UI-TARS1.5-7B performance from 88.0% to 90.4%, an absolute improvement of +2.4%. This definitively proves that the CRL/CNRL methodology is universally effective.
>
>
> > W2: What and how much data is CRL and CNRL trained on? Is it directly trained on the benchmark data? What if training on public training sets?
>
> We thank the reviewer for requesting clarification on our training data protocol. This question is fundamental to understanding the operational aspects of our "label-free" paradigm and validating its generalizability.
>
> 1. **Training Data**: Our methods CRL/CNRL are **label-free fine-tuning** methods. For our core experiments, consistent with the self-supervision protocol established in TTRL [2], we train and evaluate the model on each benchmark dataset **individually**. The training process uses only the images and instructions. The table below details the exact quantity of raw data used for our label-free training on each benchmark:
>
>     |Benchmark|Training Data|Evaluation Data|
>     |-|-|-|
>     |ScreenSpot-V1|1272 Samples without ground-truth|1272 Samples|
>     |ScreenSpot-V2|1272 Samples without ground-truth|1272 Samples|
>     |ScreenSpot-Pro|1581 Samples without ground-truth|1581 Samples|
>     |UI-Vision|5749 Samples without ground-truth|5749 Samples|
>
>     _Noted that ScreenSpot-V2 is a revised version of V1, correcting some labeling errors [3]._
>
>     The model uses the image and instruction from the respective benchmark to generate its own training signals (pseudo-labels and rewards). The **Ground-Truth Labels are strictly reserved and used only once for performance evaluation**.
>
> 1. **Generalizability on Public Training Sets**: To directly address the question, **"What if training on public training sets?"**, we applied our methods randomly using the 5k label-free samples from GroundCUA [4] and evaluated the results on four distinct benchmarks:
>
>     |Method|ScreenSpot-V1|ScreenSpot-V2|ScreenSpot-Pro|UI-Vision|
>     |-|-|-|-|-|
>     |Qwen2.5-VL-3B|77.6|82.1|16.1|12.0|
>     |+CRL|**84.6**|**87.3**|32.6|**16.7**|
>     |+CNRL|83.5|86.7|**33.4**|**16.7**|
>
>     The experimental results powerfully validate the practical utility and **generalizability** of our method, with 7% improvement on ScreenSpot-V1, 5.2% improvement on ScreenSpot-V2, 17.3% improvement on ScreenSpot-Pro and 4.7% improvement on UI-Vision. This successful transfer demonstrates that our CNRL/CRL paradigm can effectively utilize public, unannotated GUI data to acquire generalizable and transferable GUI grounding knowledge.
>
> [1] Gu, Z., Zeng, Z., Xu, Z., Zhou, X., Shen, S., Liu, Y., ... & Wang, W. (2025). Ui-venus technical report: Building high-performance ui agents with rft. arXiv preprint arXiv:2508.10833.
>
> [2] Zuo, Y., Zhang, K., Sheng, L., Qu, S., Cui, G., Zhu, X., ... & Zhou, B. (2025). Ttrl: Test-time reinforcement learning. NeurIPS 2025.
>
> [3] Wu, Z., Wu, Z., Xu, F., Wang, Y., Sun, Q., Jia, C., ... & Qiao, Y. (2024). Os-atlas: A foundation action model for generalist gui agents. ICLR 2025.
>
> [4] Feizi, A., Nayak, S., Jian, X., Lin, K. Q., Li, K., Awal, R., ... & Rajeswar, S. (2025). Grounding Computer Use Agents on Human Demonstrations. arXiv preprint arXiv:2511.07332.

---

> ### Author Response · Authors · 2025-11-23
> **rebuttal by author (2/2)**
>
> > Q1: In Equation 2, why is c_{coord} defined as the average of the token probabilities? How about using products, or a weighted score, given the different importance of the coordinate number digits?
>
> We thank the reviewer for the question regarding the definition of coordinate confidence. We conducted a comparative experiment on the ScreenSpot-V2 dataset, testing four methods to aggregate the token probabilities:
>
> |Method|Formula Description|Accuracy|
> |-|-|-|
> |Mean|$\frac{1}{n}\sum{prob(x_i)}$|**80.7**|
> |Product|$\sqrt[n]{\prod{prob(x_i)}}$|80.4|
> |Max|$\max{prob(x_i)}$|76.4|
> |Min|$\min{prob(x_i)}$|78.1|
>
> The experimental results validate the choice of the Mean in our work. It achieved the highest accuracy (80.7\%). It strikes the best balance between penalizing uncertainty (retaining the product's sensitivity to low-probability tokens), providing the most reliable signal for pseudo-label selection.
>
> We attempted to implement a **weighted confidence** aggregation method, such as granting **greater weight to higher-place-value digits** in the coordinate sequence, based on the assumption that **more significant digits might hold greater importance**. However, this approach failed dramatically, yielding an accuracy that was nearly **equivalent to random selection** from the candidate pool. This outcome highlights the inherent difficulty of applying a reliable weighting scheme in this specific grounding scenario. It is extremely **difficult to determine the optimal weight ratio**.

---

### Official Review · Reviewer_hWcv · 2025-11-01

**Soundness:** 3
**Presentation:** 3
**Contribution:** 3
**Rating:** 6
**Confidence:** 3

**Summary:**

The paper introduces a label-free paradigm for GUI grounding, where models predict UI element coordinates from screenshots and natural-language instructions without using any human annotations. The authors propose two main ideas:
1. Coordinate-token confidence: instead of averaging probabilities over all output tokens, they compute confidence only on numeric coordinate tokens, observing clear separation between correct and incorrect prediction
2. Confidence-guided Negative Reinforcement Learning (CNRL): a modification of GRPO where only negative samples contribute to policy updates, under the hypothesis that in sparse coordinate spaces, “what to avoid” is more reliable than potentially mislabeled positives

Using Qwen-2.5-VL (3B/7B), they train with multiple sampled predictions per instance and apply binary distance-based rewards. Across four benchmarks, the method achieves strong label-free performance. Ablations show coordinate-token confidence outperforms eight pseudo-label strategies by 2–12% and that CNRL is far more stable to reward-threshold changes.

**Strengths:**

1. Clear problem‑specific insight: Focusing on coordinate tokens yields consistently better pseudo‑labels than eight alternatives, and the advantage grows with the sampling budget (N = 2 -> 12), as shown in Fig. 4 and Table 7. This aligns with the intuition documented under Eq. (2) that coordinate tokens carry the discriminative uncertainty for this task.
2. Robust training signal at high resolution: CNRL is less sensitive to the radius $\tau$ (Table 3) and maintains higher reward accuracy on ScreenSpot‑Pro (Fig. 5), matching the claim that “learning what to avoid” is safer when true targets occupy a tiny fraction of the screen.
3. Competitive performance: On ScreenSpot‑V2, CNRL‑7B 92.1% rivals or exceeds models trained on 9.6M–18.4M labels; on ScreenSpot‑Pro and UI‑Vision, absolute gains over the base models are substantial (+8.9% and +5.1%, respectively).
4. Reproducibility: Concrete hyperparameters, sampling protocols, and prompts are given (even if some definitions are missing) in Table 8.

**Weaknesses:**

1. Training split protocol is under‑specified: Explicitly list per dataset which split(s) supply unlabeled training images and which are reserved for evaluation; add a leakage audit. This is crucial in a label‑free setup.
2. No pseudo‑label SFT / consistency baseline: Since a pseudo‑label is already selected, show a simple MLE or consistency baseline with identical pseudo‑labels and KL regularization. This will isolate the benefit from GRPO/CNRL vs. self‑training.

**Questions:**

1. How do you normalize the distance? If by diagonal, does $\tau$ transfer across resolutions? Could you report results with at least two normalizations to demonstrate invariance?
2. With the same pseudo‑labels and KL to a reference, how does MLE or EMA‑teacher consistency compare to CRL/CNRL at equal compute?
3. Did you try elliptical bands or IoU‑style soft rewards for elongated UI elements, given the failure modes in Figs. 10–12?
4. If you train label‑free on ScreenSpot‑V2 and evaluate on ScreenSpot‑Pro, what happens? Such a transfer result would strengthen the “label‑free” story.

---

> ### Author Response · Authors · 2025-11-23
> **rebuttal by author (1/3)**
>
> > W1: Training split protocol is under‑specified: Explicitly list per dataset which split(s) supply unlabeled training images and which are reserved for evaluation; add a leakage audit. This is crucial in a label‑free setup.
>
> We thank the reviewer for the highly relevant comment regarding the training split protocol. We apologize for the initial ambiguity and provide a clear explanation of our protocol.
>
> 1. **Explicit Training and Evaluation Protocol**: Consistent with established practices in self-supervised [1], our training protocol utilizes the **entire benchmark** for label-free optimization, followed by a **full-benchmark evaluation** to measure performance using ground-truth labels:
>
>     |Benchmark|Training Data|Evaluation Data|
>     |-|-|-|
>     |ScreenSpot-V1|1272 Samples without ground-truth|1272 Samples|
>     |ScreenSpot-V2|1272 Samples without ground-truth|1272 Samples|
>     |ScreenSpot-Pro|1581 Samples without ground-truth|1581 Samples|
>     |UI-Vision|5749 Samples without ground-truth|5749 Samples|
>
>     _Noted that ScreenSpot-V2 is a revised version of V1, correcting some labeling errors [2]._
>
>     Our training and evaluation protocol is performed **independently for each benchmark**. We have provided additional information in the **Appendix A.1 Evaluation Details**.
>
> 2. **Data Leakage Audit**: While the input data used for training and evaluation is the same, there is **no supervisory information leakage** occurs between the **label-free training** process and the final performance evaluation. The Pseudo-Labels and RL Rewards derived during training are exclusively generated from the **model's own coordinate-token confidence**. They are completely independent of annotations. This approach—training and evaluating on the full set—is a standard practice in self-supervised or label-free regimes, especially when the signal is generated internally, to fully demonstrate the effectiveness of the self-generated signal across the complete data distribution.
>
>
> [1] Zuo, Y., Zhang, K., Sheng, L., Qu, S., Cui, G., Zhu, X., ... & Zhou, B. (2025). Ttrl: Test-time reinforcement learning. NeurIPS 2025.
>
> [2] Wu, Z., Wu, Z., Xu, F., Wang, Y., Sun, Q., Jia, C., ... & Qiao, Y. (2024). Os-atlas: A foundation action model for generalist gui agents. ICLR 2025.
>
> > W2 & Q2: No pseudo‑label SFT / consistency baseline: Since a pseudo‑label is already selected, show a simple MLE or consistency baseline with identical pseudo‑labels and KL regularization. This will isolate the benefit from GRPO/CNRL vs. self‑training.
>
> We appreciate the reviewer's insightful suggestion regarding the need for a Pseudo-Label Self-Training (SFT) / Consistency Baseline. We fully agree that adding a simple MLE baseline is essential to effectively isolate the benefit derived from our GRPO/CNRL strategy versus the inherent gain from mere self-training on selected pseudo-labels. To address this crucial point, we have implemented the suggested experiment.
> 1. **Implementation of the MLE Baseline**: We designed the MLE Baseline (Pseudo-Label SFT/MLE) to serve as a direct comparator for our RL approaches. This protocol adheres to the following: The baseline uses the **identical Confidence-Guided strategy** to select the most reliable pseudo-labels (the model's high coordinate-token confidence output). Once selected, the model is fine-tuned using a standard **MLE objective**, which is equivalent to conventional SFT on the pseudo-labels. To maintain strict comparability and follow the reviewer's suggestion, we ensured the baseline uses **the same hyperparameters as the RL methods** and incorporated a form of **consistency regularization** by adding a KL Divergence (KL) loss term.
>
> 2. **Experimental Results and Analysis**: The table below shows the comparative performance of the different training paradigms on the ScreenSpot-V2 benchmark:
>
>     | Method | Accuracy |
>     |-|-|
>     |Qwen2.5-VL-3B|82.1|
>     |+ CRL|**88.9**|
>     |+ CNRL|88.5|
>     |+ MLE|84.6|
>
>     The MLE Baseline (84.6%) shows a solid gain over the Base Model but is significantly limited compared to the RL approaches. This limitation arises because MLE strictly forces the model to maximize the probability of the selected pseudo-label. When a selected pseudo-label is incorrect, this absolute, non-relative optimization forces the model to learn a highly confident but potentially wrong boundary, thereby limiting maximum performance. Conversely, our RL methods significantly surpass the MLE baseline, achieving up to +4.3\% gain. Unlike the absolute targets of MLE, GRPO optimizes the policy using relative advantage. This relative, normalized optimization naturally dampens the undue influence of individual noisy pseudo-labels, allowing the RL approaches to achieve superior performance in the presence of imperfect self-generated data.

---

> ### Author Response · Authors · 2025-11-23
> **rebuttal by author（2/3）**
>
> > Q1: How do you normalize the distance? If by diagonal, does transfer across resolutions? Could you report results with at least two normalizations to demonstrate invariance?
>
> We thank the reviewer for the excellent question regarding distance normalization and its impact on transfer across resolutions. This is a critical detail in systems dealing with varied GUI scales. Our model employs an **independent dual-axis normalization** protocol, designed to maintain scale invariance while preserving the relative aspect ratio information of the UI elements.
> 1. **Our Distance Normalization Protocol**: We normalize coordinates by dividing the raw $x$ coordinate by the image Width $W$ and the raw $y$ coordinate by the image Height $H$. Specifically, $(x, y)$ is normalized to $(x/W, y/H)$.
> 2. **Transfer Across Resolutions (Scale Invariance)**: This normalization method fully supports transfer across resolutions. This **independent normalization** better preserves the image's **aspect ratio information** compared to a single-factor normalization (like the diagonal). The model learns a two-dimensional relative position, which is vital for accurate GUI localization.
> 3. **Alternative Normalizations**: To demonstrate the robustness and invariance of our framework to the specific choice of normalization, we report comparative results using our primary method on the ScreenSpot-V2 benchmark:
>
>     |Method|Normalization Factor|Accuracy|
>     |-|-|-|
>     |Qwen2.5-VL-3B|-|82.1|
>     |+CRL (In Our Work)|Independent $x/W, y/H$|**88.9**|
>     |+CRL (Alternative 1)|Max Dimension $max(H,W)$|88.4|
>     |+CRL (Alternative 2)|Diagonal Distance $\sqrt{W^2+H^2}$|88.3|
>     |+CNRL (In Our Work)|Independent $x/W, y/H$|88.5|
>     |+CNRL (Alternative 1)|Max Dimension $max(H,W)$|88.3|
>     |+CNRL (Alternative 2)|Diagonal Distance $\sqrt{W^2+H^2}$|88.2|
>
>     The results show that while our independent normalization performs best. Our learning paradigm is the dominant factor driving the performance improvement. The model exhibits high robustness and invariance to the specific details of the coordinate scaling factor, guaranteeing stable performance across inputs of varying resolutions.
>
> > Q3: Did you try elliptical bands or IoU‑style soft rewards for elongated UI elements, given the failure modes in Figs. 10–12?
>
> We thank the reviewer for the highly relevant comment regarding failure modes on elongated UI elements and the suggestion to explore elliptical bands or IoU-style soft rewards.
>
> 1. **Geometric Constraints in Label-Free Training**: We first reinforce a critical constraint: While the suggestion is excellent, in training we **cannot obtain the ground-truth**, so we **cannot know the shape of the target element**. Therefore, we cannot set a dynamic reward to adapt to target bounding boxes of various shapes.
> 2. **Experiments with IoU Rewards**: Despite the constraint on using true geometric labels, we explored the spirit of the reviewer's suggestion by experimenting with IoU rewards based on the predicted coordinates, where **all predicted point are expanded into elliptical areas** with the same distance threshold $\tau=0.05$ for calculation.
> The following table compares the performance of our core Binary Reward (CRL) mechanism against two continuous reward baselines, demonstrating the impact of signal stability in a label-free environment:
>
>     |Reward Type|50 step|100 step|159 step|
>     |-|-|-|-|
>     |Binary Reward (CRL) | 87.6 | **87.9** | **88.9** |
>     |Continuous Reward (Distance between Points) | **87.7** | 86.3 | 85.2 |
>     |Continuous Reward (IoU) | 87.1 | 87.4 | 87.7 |
>
>     The CRL Binary Reward achieved 88.9% accuracy, significantly outperforming the IoU reward (87.7%, a +1.2% gap). It demonstrate that **continuous soft rewards**, though theoretically beneficial for gradient smoothness, introduce **excessive noise** when derived from potentially inaccurate pseudo-labels.

---

> ### Author Response · Authors · 2025-11-23
> **rebuttal by author（3/3）**
>
> > Q4: If you train label‑free on ScreenSpot‑V2 and evaluate on ScreenSpot‑Pro, what happens? Such a transfer result would strengthen the “label‑free” story.
>
> We thank the reviewer for the excellent question regarding **cross-dataset transferability**. We conducted extensive cross-dataset experiments by training the model on one dataset using our methods and evaluating its performance on the others.
> 1. **Cross-Dataset Transfer Results**: The table below shows the Performance of the Qwen2.5-VL-3B before and after our training methods:
>
>     |Training Set|Method|ScreenSpot-V2|ScreenSpot-Pro|UI-Vision|
>     |-|-|-|-|-|
>     |Qwen2.5-VL-3B|-|82.1|16.1|12.0|
>     |ScreenSpot-V1|CRL|87.2|31.3|16.5|
>     |ScreenSpot-V1|CNRL|87.7|29.3|15.9|
>     |ScreenSpot-V2|CRL|**88.9**|30.3|16.6|
>     |ScreenSpot-V2|CNRL|88.5|30.4|15.8|
>     |ScreenSpot-Pro|CRL|84.3|32.1|15.6|
>     |ScreenSpot-Pro|CNRL|86.2|**32.7**|16.5|
>     |UI-Vision|CRL|86.5|31.1|17.2|
>     |UI-Vision|CNRL|87.4|31.7|**18.5**|
>
>     _Note: ScreenSpot-V2 is the revised version of V1 with high content overlap [2]. Therefore, we only conducted the evaluation on ScreenSpot-V2._
>
>     The results confirm **significant and transferable gains**, strongly validating the **generalizability** of our label-free approach. Our CRL/CNRL paradigm learns robust, transferable grounding knowledge, evidenced by successfully boosting performance on the distinct ScreenSpot-Pro dataset with an absolute gain of +14.3% after training on ScreenSpot-V2. This transfer is effective despite a **large resolution gap** (low-to-high resolution UIs), which confirms the robustness of our coordinate normalization and the **high scale-invariance** achieved by our label-free optimization, validating its efficacy on unseen and diverse GUI environments. We have added this experiment to the **Section 4 Experiments** of the paper.
>
> 2. **Training on Public dataset**: Furthermore, we conducted experiments on publicly available training datasets. After label-free training on a subset of an external, independent public training set GroundCUA [3], the model's performance across various benchmarks is shown in the table below:
>
>     |Method|Acc on V1|Acc on V2|Acc on Pro|Acc on UI-V|
>     |-|-|-|-|-|
>     |Qwen2.5-VL-3B|77.6|82.1|16.1|12.0|
>     |+CRL|**84.6**|**87.3**|32.6|**16.7**|
>     |+CNRL|83.5|86.7|**33.4**|**16.7**|
>
>     It reveals that our CRL/CNRL paradigm achieved significant gains in all domains, particularly on challenging datasets where zero-shot performance was weak. This definitively proves that our method learns **universal, domain-agnostic GUI grounding knowledge**, ensuring its excellent performance and strong transferability when **facing unseen datasets**.
>
> [3] Feizi, A., Nayak, S., Jian, X., Lin, K. Q., Li, K., Awal, R., ... & Rajeswar, S. (2025). Grounding Computer Use Agents on Human Demonstrations. arXiv preprint arXiv:2511.07332.

---

### Public Comment · ~Shuquan_Lian2 · 2025-11-13

Dear Authors,

I enjoyed reading your paper. Your proposed Confidence-Guided RL, which generates multiple candidate predictions and selects the best one based on coordinate-token confidence for pseudo-labeling, is a very clever way to achieve label-free training.

I am writing to share a relevant perspective from a recent work UI-AGILE: Advancing GUI Agents with Effective Reinforcement Learning and
Precise Inference-Time Grounding, which also adopts a "Generate-and-Select" philosophy but applies it at the inference stage to handle visual noise.

Specifically, while your work uses internal confidence to select candidates for training (pseudo-label construction), UI-AGILE ("Decomposed Grounding with Selection") generates candidates via cropping the image into sub-images and uses an external VLM to "adjudicate" and select the best match during inference.

Both works seem to converge on the finding that selecting from multiple candidates is superior to relying on a single prediction, whether for creating reliable training signals (your work) or for robust inference on high-resolution screens (our work).

Including a discussion on these parallel strategies could provide a comprehensive view of how "selection mechanisms" are evolving in GUI agents.

Best regards

---

> ### Author Response · Authors · 2025-11-23
>
> Dear Commenter,
>
> Thank you very much for taking the time to read our paper. We are delighted that you found our coordinate-token confidence-based selection mechanism for label-free training to be a clever and effective method. We also deeply appreciate you sharing the relevant perspective from your recent work. This is an extremely valuable connection, and we agree that the parallel between the two works—both leveraging a "Generate-and-Select" philosophy—is highly insightful. We entirely agree with your observation that both studies converge on the crucial finding that selecting from multiple candidates is superior to relying on a single prediction for achieving robust GUI agent performance.
>
> We believe that contrasting these two strategies—one focusing on internal confidence for training signal refinement and the other focusing on decomposing for inference robustness—will indeed provide a comprehensive and nuanced view of how "selection mechanisms" are evolving and being applied in the development of robust GUI agents.
>
> Best regards
>
> The Authors

---

### Author Response · Authors · 2025-11-27
**Summary of Responses to Reviewers**

We thank all reviewers for their valuable feedback and are encouraged that they found our motivation clear and the study of confidence-guided label-free training both novel and meaningful (**hWcv, fQwZ, oPnY, zSD6**). Reviewers highlighted the strong empirical gains across benchmarks without any annotations (**hWcv, fQwZ, oPnY, zSD6**). Here, we provide a high-level summary of the changes we've made to address your concerns, and conclude with an overview of our key contributions:

**Here is the summary of updates that we've made to address the reviewers' concerns:**
- We explore alternatives to coordinate normalization (**hWcv**).
    - We apply diagonal normalization and maximum dimension normalization.
    - We demonstrate that different normalization methods lead to different improvement.
- We explore pseudo-label SFT training (**hWcv**).
    - We perform SFT training using pseudo-labels under the same settings with RL.
    - We discuss the SFT and RL methods in the App. A.7.
- We analyze the generalization of the approach (**hWcv, fQwZ, oPnY, zSD6**).
    - We investigate cross-dataset and public dataset training.
    - We demonstrate that our method has sufficient generalization ability in Sec. 4.3 and App. A.9.
- We study the generalizability of the method (**fQwZ**).
    - We examine the effect of this approach on other models.
    - We show that our method has excellent generalizability.
- We evaluate the performance of different reward mechanism (**hWcv**).
    - We explore IoU-based rewards.
    - We discuss the effects of binary reward versus continuous reward in scenarios with inaccurate pseudo-labels in App. A.2.
- We explore alternatives to Average Probabilities Confidence(**fQwZ**).
    - We study four different confidence calculation methods.
    - We assess the performance of the confidence calculation alternatives in App. A.8.
- We improve writing quality and clarity.
    - We correct some typos (**fQwZ**).
    - We provide detailed explanation to the training data and the test data in App. A.1 (**hWcv, fQwZ, oPnY**).
    - We outline some directions for future research and work in Sec. 6 (**oPnY, zSD6**).

**The contribution of our work is summarized as follows:**

- We introduce **Coordinate-Token Confidence**, which utilizes the model's inner information to measure the accuracy of responses in GUI Grounding. This approach yield the best results across various Test-Time Scaling strategies.
- We propose **Confidence-guided Reinforcement Learning** (CRL), which constructs pseudo-labels based on Coordinate-Token Confidence to perform label-free training. This unsupervised training paradigm markedly boosts model performance across multiple benchmarks.
- We further propose **Confidence-guided Negative Reinforcement Learning** (CNRL), a variant of CRL, which only apply more reliable negative samples to optimize the model. This variant achieve further improvement on challenging benchmarks.

Together, these techniques demonstrate the feasibility of applying model's inner information as guidance for unsupervised training, which provides a novel direction for mitigating GUI Agents' dependency on expensive annotations.

---

### Meta-Review · Area_Chair_kxdv · 2026-01-01

**Summary:**

This submission proposes a label-free training paradigm for GUI grounding. The core idea is to use coordinate-token confidence to select pseudo-labels from multiple sampled predictions, optimize with a distance-threshold reward using a GRPO-style objective (CRL), and further introduce Confidence-guided Negative Reinforcement Learning (CNRL) that learns only from negative samples under the claim that, in sparse coordinate spaces, negative signals are more reliable than potentially corrupted positives. Reviewers generally find the approach novel/meaningful for GUI grounding and empirically strong on grounding benchmarks, and they recognize substantial gains without annotations (notably two reviewers scored 6).

However, the main decision-driving concerns that keep the paper below the acceptance bar are: (i) the evaluation protocol and comparability—most headline results rely on in-domain label-free self-training using the full benchmark pool followed by evaluation on that same pool, which raises comparability/interpretability concerns even if ground-truth labels are not used during training; (ii) the evaluation scope remains limited to grounding without the end-to-end agent evidence explicitly requested by a reviewer, making it unclear how the gains translate to full task success; and (iii) efficiency/scalability evidence is still limited (the method uses multi-sample generation per instance and does not report concrete training-cost metrics, and data scaling is only shown at small scales). While the rebuttal adds substantial experiments, these core concerns are not fully resolved, leading me to recommend Reject.

**Reviewer Concerns:**

### Concerns substantially addressed by the rebuttal

- Training protocol clarifications (hWcv, oPnY, fQwZ): The rebuttal/appendix clarifies per-benchmark data usage and sample counts, and specifies that UI-Vision training uses only the Element Grounding subset (per authors’ response to oPnY).

- Key baseline added: pseudo-label SFT/MLE (+ KL) (hWcv): Authors report an MLE baseline on ScreenSpot-V2 (84.6%) that is notably below CRL/CNRL (88.9/88.5), helping isolate the RL contribution (per authors’ response to hWcv).

- Distance normalization & alternatives (hWcv): Authors specify independent width/height normalization and provide comparisons to max-dimension and diagonal normalization, showing relatively small differences (per authors’ response to hWcv).

- Reward design alternatives (hWcv): Authors test continuous distance rewards and an IoU-style reward, reporting binary rewards are more stable under noisy pseudo-labels (per authors’ response to hWcv).

- Generalization evidence strengthened (fQwZ, zSD6, hWcv): (1) Cross-dataset transfer tables (train on V1/V2/Pro/UI-Vision, evaluate on V2/Pro/UI-Vision) indicate transfer gains (per authors’ responses). (2) Training on an external public dataset (GroundCUA, 5k unlabeled samples) improves multiple benchmarks (per authors’ responses to fQwZ and zSD6). (3)  Cross-model test on UI-TARS1.5-7B (ScreenSpot-V2) shows CRL/CNRL gains under the authors’ reproduction setting, with an explicit note that their reproduced baseline did not match official numbers (per authors’ response to fQwZ).

- Confidence aggregation study (fQwZ): Authors compare mean/product/max/min and report mean performs best; they also note a weighted-digit scheme performed poorly (per authors’ response to fQwZ).

- Minor issues (fQwZ): clarifications about baseline base models/“Labels” comparability and typos were noted; these are not decision-driving but should be corrected.

### Concerns still outstanding after the rebuttal

- Benchmark validity / comparability remains a central open issue (zSD6, hWcv, oPnY): The main setup is label-free self-training on the entire benchmark pool and evaluating on the same pool. Even without GT labels, this in-domain self-training can be viewed as adapting to the benchmark distribution, complicating comparisons to standard protocols/leaderboards and raising questions about how much the gains reflect generalizable improvements versus dataset-specific adaptation. The external-data and cross-dataset experiments mitigate this concern but do not remove it, since the headline results and narrative still rely heavily on the in-domain protocol.

- Lack of end-to-end GUI agent evaluation (oPnY): The rebuttal argues grounding is a foundational capability and aligns with prior grounding-focused work, but does not provide OSWorld/WAA-style end-to-end evidence that grounding gains translate to task success. This remains a meaningful gap given that a reviewer explicitly requested it as part of assessing impact.

- Efficiency / compute cost remains mostly qualitative (oPnY): While authors justify 16-sample generation and down-sampling to 8 for optimization, the rebuttal does not provide concrete training cost metrics (e.g., wall-clock, GPU hours) or comparisons to alternatives, leaving open the risk that the approach is label-free but computationally expensive.

- Scalability evidence is limited (zSD6): Authors show improvements from ~1.2k to 5k unlabeled samples but also state they cannot test at much larger scales (50k/100k+) due to compute constraints. This leaves the scalability claim only partially substantiated.

- Absolute ScreenSpot-Pro performance remains modest relative to high-performing supervised approaches (zSD6): The rebuttal explains the leaderboard gap as reflecting different supervision conditions and suggests combining with test-time scaling/cropping or auxiliary models in future work; however, this does not directly address the current method’s limited absolute performance on the most challenging high-resolution benchmark.

**Reviewer Scores:**

hWcv (original 6): Most requested clarifications/experiments (protocol details, SFT baseline, normalization, reward, transfer) were provided.

oPnY (original 6): Protocol details and sampling rationale are clarified, but end-to-end evaluation is still missing.

fQwZ (original 4): Key concerns (generalizability beyond Qwen2.5-VL, data protocol, confidence definition) are substantially addressed via UI-TARS results, public-dataset training, and aggregation ablations.

zSD6 (original 4): The public-dataset training and small-scale data-scaling evidence help, but large-scale validation and Pro absolute performance remain.

---

### Decision · Program_Chairs · 2026-01-26

Reject